# Advances in Research on Type 2 Diabetes Mellitus Targets and Therapeutic Agents

**DOI:** 10.3390/ijms241713381

**Published:** 2023-08-29

**Authors:** Jingqian Su, Yingsheng Luo, Shan Hu, Lu Tang, Songying Ouyang

**Affiliations:** 1Fujian Key Laboratory of Innate Immune Biology, Biomedical Research Center of South China, Fujian Normal University, Fuzhou 350117, China; sjq027@fjnu.edu.cn (J.S.); yingshengluo_77@163.com (Y.L.); qbx20220152@yjs.fjnu.edu.cn (S.H.); qsx20211378@student.fjnu.edu.cn (L.T.); 2Provincial University Key Laboratory of Microbial Pathogenesis and Interventions, Fujian Normal University, Fuzhou 350117, China; 3Provincial University Key Laboratory of Cellular Stress Response and Metabolic Regulation, Fujian Normal University, Fuzhou 350117, China; 4Key Laboratory of OptoElectronic Science and Technology for Medicine of the Ministry of Education, College of Life Sciences, Fujian Normal University, Fuzhou 350117, China

**Keywords:** type 2 diabetes mellitus, target, mechanism, medicine

## Abstract

Diabetes mellitus is a chronic multifaceted disease with multiple potential complications, the treatment of which can only delay and prolong the terminal stage of the disease, i.e., type 2 diabetes mellitus (T2DM). The World Health Organization predicts that diabetes will be the seventh leading cause of death by 2030. Although many antidiabetic medicines have been successfully developed in recent years, such as GLP-1 receptor agonists and SGLT-2 inhibitors, single-target drugs are gradually failing to meet the therapeutic requirements owing to the individual variability, diversity of pathogenesis, and organismal resistance. Therefore, there remains a need to investigate the pathogenesis of T2DM in more depth, identify multiple therapeutic targets, and provide improved glycemic control solutions. This review presents an overview of the mechanisms of action and the development of the latest therapeutic agents targeting T2DM in recent years. It also discusses emerging target-based therapies and new potential therapeutic targets that have emerged within the last three years. The aim of our review is to provide a theoretical basis for further advancement in targeted therapies for T2DM.

## 1. Introduction

Diabetes mellitus is a chronic metabolic disease characterized by hyperglycemia. In 2022, the American Diabetes Association (ADA) categorized diabetes into four main types based on different pathogenesis: type 1 diabetes, type 2 diabetes, specific types of diabetes caused by other reasons (e.g., monogenic diabetes syndromes, diseases of the exocrine pancreas, and drug- or chemically induced diabetes), and gestational diabetes, with type 2 diabetes accounting for 90–95% of all cases [1]. Type 2 diabetes mellitus (T2DM), also known as non-insulin-dependent diabetes mellitus, is polygenic and multifactorial in its pathogenesis, with genetics, lifestyle habits, and the acquired health status of the organism influencing the development of the disease. Newly confirmed models of pathogenesis suggest that nutritional loading causes a chronic increase in insulin secretion, leading to hyperinsulinemia, which in turn triggers insulin resistance until β-cell failure, ultimately challenging the conventional belief that insulin resistance precedes the onset of hyperinsulinemia in the development of overt T2DM [2]. According to the latest data released by the International Diabetes Federation (https://diabetesatlas.org/ (accessed on 22 June 2023)), 537 million people (more than 75% of whom are from low- and middle-income countries) were diagnosed with diabetes in 2021, which has been expected to increase to 643 million by 2030 and to 783 million by 2045. In terms of mortality, this disease was expected to cause 6.7 million deaths in 2021 alone, equivalating to one death every 5 s. Patients with diabetes are also highly susceptible to complications, with cancers, such as gastrointestinal, pancreatic, and ovarian cancer, and dementia, such as Alzheimer’s disease, becoming the leading causes of death for people with diabetes in some regions; further, patients with diabetes are at a high risk of infections, primarily including foot, respiratory, urinary tract, and postoperative infections [3]. In the context of the COVID-19 pandemic, hyperglycemia in patients with diabetes increases SARS-CoV-2 replication [4], and some anti-glycemic drugs may upregulate ACE2 expression levels, all of which increase the susceptibility to and severity of COVID-19 [5]. In addition, diabetes causes cognitive dysfunction, affective disorders, obstructive sleep apnea, and liver disease; furthermore, there is evidence of a mutual exacerbation between these conditions and the underlying disease [3]. Therefore, the effective treatment of patients with diabetes remains warranted.

However, to date, the cure for diabetes is lacking, while treatments, such as medications, are available to maintain the blood glucose level as close to normal levels as possible, thus delaying or preventing the occurrence of diabetes-related health problems. Further, owing to drug resistance, long-term treatment limited to a particular drug against a specific target may gradually become ineffective. Different individuals may have different underlying diseases, and treatment against a certain target may not be applicable to everyone due to their variability. In addition, multi-target drugs are more effective than single-target drugs. Therefore, there is a need to continuously identify new targets and develop efficient and safe therapeutic drugs. Meanwhile, owing to the cumbersome process of long-term drug administration, emerging technologies, such as stem cell therapy and CRISPR therapy, can be attempted to resolve such difficulties. Accordingly, this review presents an overview of the mechanisms of action and the development of the latest therapeutic agents targeting T2DM in recent years. It also discusses emerging target-based therapies and new potential therapeutic targets that have emerged within the last three years. The aim is to provide a theoretical basis for further advancement in targeted therapies for T2DM.

## 2. Methods

PubMed, Scopus, Web of Science, and Google Scholar databases were used as sources, and the terms “target, drug, mechanism, diabetes mellitus, and insulin resistance” were used as the main keywords to collect the relevant literature from 2018–2023 for analysis, and, based on the specifics of the reviewed targets as the keywords, the relevant literature from the past 3 years was collected for discussion.

## 3. Incretin-Based Targets and Therapeutic Medicines

### 3.1. Glucagon-like Peptide-1 (GLP-1) and Glucose-Dependent Insulinotropic Polypeptide (GIP)

Both GLP-1 and GIP are members of the incretin family; GLP-1 is released by L cells in the distal ileum and colon, while GIP is released via K cells in the duodenum and jejunum [6]. Incretin is a gastrointestinal hormone released after consuming nutrients and upon its release, activates receptors on pancreatic β-cells, thereby enhancing insulin secretion and digestion and catabolism of carbohydrates and lipids in a glucose concentration-dependent manner [7]. However, the specific mechanisms of blood glucose regulation differ between GLP-1 and GIP (Figure 1). GLP-1 inhibits glucagon secretion along with stimulating insulin secretion during hyperglycemia [8], reducing appetite, delaying gastric emptying by stimulating vagal afferents, activating hindbrain PKA and MAPK, and inhibiting hindbrain 5’-AMPK phosphorylation [9,10]. GIP is a gastric peptide [11] that inhibits gastric acid and pepsinogen secretion, thereby delaying gastric motility [12], but has no effect on the inhibition of glucagon secretion. It also enhances glucagon secretion during hypoglycemia [8] and has a direct effect on lipid homeostasis [13].

Drugs that target GLP-1 are mainly its receptor agonists; these drugs have replaced insulin as the main line of treatment for diabetes. The efficacy of GLP-1 receptor agonists is similar to that of bariatric surgery (gastrectomy and gastric bypass), which exacerbates the release of incretin, with GLP-1 levels increasing up to 10-fold or more, displaying the range of concentrations typically achieved during agonist therapy [14]. However, agonists developed in the early stage were found to have had a short half-life. Subsequently, the strategy of coupling them with free fatty acid side chains or large proteins (such as immunoglobulins) has improved this defect [15], and a long-acting drug that can better reduce glycated hemoglobin (HbA1c) has been developed. The current GLP-1 single-target drug is the long-acting drug semaglutide.

GIP receptor agonists are not used as single-target drugs to treat patients with diabetes, because after GIPR is activated, in addition to the lipogenic effect of insulin, GIP can recruit chylomicrons, enhance the activity of lipoprotein lipase, promote the hydrolysis of dietary triglycerides and the release of free fatty acids, and promote lipid storage, thus leading to obesity [7]. In addition, obesity can aggravate the condition of patients with T2DM [12]. However, the combination of GIPR with GLP-1R showed a superior effect on blood glucose and weight reduction than GLP-1R single agonists, and the addition of GIP also reduced the incidence of gastrointestinal events prevalent with GLP-1R agonists. Data from three preclinical animal studies (mice, rats, and musk shrews) showed that GIPR activation attenuated the complications of heterophagia and conditioned taste avoidance caused by GLP-1R activation, while significantly improving glucose tolerance, suppressing weight gain, and reducing appetite. Further, G-aminobutyric acidergic neurons expressed in the hindbrain may act as local inhibitory neurons that mediate the antiemetic effects of GIPR signaling [16].

Tirzepatide was the first US Food and Drug Administration (FDA)-approved GLP-1 and GIP receptor dual agonist drug. The results of the phase III clinical trial of tirzepatide showed a significant reduction in HbA1c levels and a more pronounced weight loss in the experimental group compared with those in the placebo group, and the level of reduction was determined to be dose-dependent [13]. A meta-analysis also revealed that tirzepatide is more effective than the placebo and less prone to hypoglycemic events [6]. Furthermore, tirzepatide may prevent cardiovascular disease and atherosclerosis, among others [17,18]. However, existing data appear to contradict the previous results that GIP can reduce the incidence of gastrointestinal events caused by GLP-1R agonists. Instead, the incidence of gastrointestinal events increase following treatment with tirzepatide, possibly owing to the suboptimal dose implemented in clinical trials [16]. However, it does not cause serious adverse events nor death regardless of the dose administered [6]. Only 20–30% of the insulin sensitization effect of tirzepatide leads to weight loss, but GLP-1R agonists are entirely attributable to weight loss, which indicates that tirzepatide can more directly affect pancreatic cells and insulin sensitivity [7,19].

### 3.2. Dipeptidyl Peptidase-4 (DPP-4)

DPP-4, also known as T-cell surface antigen CD26, is a transmembrane serine protease expressed on the cell surface found mainly in the small intestine, bile ducts, liver, kidney, pancreas, and some endothelial cells, immune cells, and fibroblasts [20,21]. DPP-4 exists in both membrane-bound and soluble forms, and soluble DPP-4 is enzymatically active. The substrates of DPP-4 cover multiple types, including growth factors, cytokines, chemokines, and neuropeptides, and regulate the biological function of the substrate by acting on the penultimate N-terminal Pro or Ala of the substrate [20]. GLP-1 and GIP are endogenous substrates of DPP-4 that are degraded to form inactive metabolites, which allows for the inhibition of their mediated hypoglycemic pathway (Figure 1). This, therefore, provides a concept for the development of a DPP- 4 inhibitor [22]. The hypoglycemic effect of DPP-4 inhibitors is also glucose dependent, involving a dual action on islet α- and β-cells, which promotes insulin release from β-cells by keeping incretin intact in the form of a bioactive peptide, while inhibiting glucagon secretion from α cells and preventing hypoglycemic risk due to the negative feedback regulation of glucagon via GIP in hypoglycemia. In addition, peptide YY (PYY) is also a substrate of DPP-4, and DPP-4 inhibitors reduce the conversion of PYY into its anorexigenic metabolites, an effect that may counteract the weight loss effect of increased GLP-1 and maintain weight neutrality [23]. The current gliptin series of drugs for DPP-4 targets are small molecules, which are cheap to manufacture, easily absorbed, highly stable, have low mutagenicity, and do not require special storage conditions. Furthermore, DPP-4 inhibitors are bound to the catalytic site of DPP-4 in a reversible manner and generally do not affect other known functions of DPP-4 independent of the enzyme [23]. Although DPP-4 inhibitors dose-dependently inhibit DNA synthesis and IgM secretion by B cells, this stimulatory effect is not necessary; thus, DPP-4 inhibition does not impair immune function [20].

Currently, there are five mainstream gliptin drugs, of which the peptides are sitagliptin, vildagliptin, and saxagliptin, and the non-peptide drugs are alogliptin and ligliptin, which differ in their metabolic pathway (hepatic/renal), half-life, and effective dose due to their structural classes [24]. The FDA has provided warnings regarding the increased risk of heart failure events with DPP-4 inhibitors, but an updated meta-analysis showed no significant risk of heart failure events [25]. When it was used in conjunction with metformin, the incidence of cardiovascular events was not found to be significantly different from that of SGLT-2 inhibitors with cardiovascular benefits [26]. The incidence of major adverse cardiovascular events was also found to be lower when compared with that of sulfonylureas with similar efficacy [27]. However, a significant correlation was uncovered between DPP-4 inhibitors and the development of immune disorders, such as herpetic pemphigoid [28].

### 3.3. G protein-Coupled Receptors (GPCRs)

GPCRs are the largest family of membrane proteins in the human genome and the most common targets for FDA-approved drugs owing to their high drug-containing properties and the multiple physiological pathways mediated in the human body [29,30]. GPCRs activated by fatty acid-derived lipids can control glucose homeostasis by affecting insulin and incretin secretion and are considered potential targets for the treatment of diabetes, including free fatty acid receptor 1 (FFA1/GPR40), FFA4 (GPR120), and lipid metabolite-binding glucose-dependent insulinotropic receptor (GPR119) [31].

GPR40 belongs to the class A GPCRs, a member of the rhodopsin-like family with a conventional seven transmembrane structural domain and an additional eight helices at the C-terminus linked to palmitoylated cysteines, which can be activated using endogenous medium- and long-chain fatty acids and their derivatives [31,32]. It is mainly distributed in pancreatic β-cells, intestinal endocrine L, K, and I cells, pancreatic α-cells, and the central nervous system [33,34]. GPR40 has two binding sites, an endogenous FFA-binding orthosteric site and a variant binding site (interhelical A1 site and extrahelical A2 site) [33], which can initiate different regulatory mechanisms (Figure 2). The orthosteric site and the A1 site are coupled as Gαq/11, and, when activated, increases intracellular Ca^2+^ levels, causing glucose-dependent insulin secretion (GSIS) [31,33,34]. The A2 site is a lipid-rich region that, when bound, stabilizes a conformation important for Gαs coupling, which increases intracellular cAMP levels, causing the secretion of enterostatin [31,34]. GPR40 also activates the β-arrestin-mediated signaling pathway, which is associated with GSIS [33]. The GPR40 agonists currently developed are divided into partial agonists and full agonists/orthoallosteric modulation (AgoPAM) agonists, with partial agonists generally activating only Gαq/11 and full agonists activating both Gαq/11 and Gαs; however, neither are available for clinical use due to either their poor efficacy or severe side effects [34]. TAK-875, the only GPR40 agonist that has entered phase III clinical trials [35], is a partial agonist bound to the A1 site and significantly reduces HbA1c levels in patients with T2DM [33,34]. However, TAK-875 was terminated at this stage owing to hepatotoxicity issues, probably due to its own metabolites, which affect bile acid and bilirubin homeostasis, leading to hyperbilirubinemia and cholestatic hepatotoxicity [33]. Further, GPR40 activation on β-cells may also trigger glucolipotoxicity, but this damage is not target-specific but rather determined by the drug itself, that is, different drugs do not necessarily trigger this adverse effect, as TAK-875 did not show β-cell toxicity in several rat experiments [34].

SCO-267 was the first full agonist to be revealed in a clinical trial profile and can be administered orally once daily. In phase I clinical trials, SCO-267 was found to be safe and well tolerated after single and multiple doses in both healthy adults and patients with diabetes, was effective in improving glucose tolerance in patients with diabetes, and did not cause hypoglycemia; it is currently being developed for phase II trials [36].

GPR119, which is a class A GPCR [32], is highly expressed in pancreatic β-cells and intestinal endocrine K and L cells activated by endogenous cannabinoid-like compounds, long-chain fatty acyl ethanolamines, and oleic acid derivatives [31,37,38]. GPR119 is mainly coupled to the Gαs pathway and activated to increase intracellular cAMP levels, thereby promoting GSIS and incretin secretion [39] (Figure 2). At the same time, GPR119 activation also increases glucagon secretion, which may reduce the risk of medically induced hypoglycemia in patients with diabetes undergoing intensive insulin therapy [40]. Currently, no GPR119 agonists have entered clinical phase III trials, possibly owing to its poor efficacy, as the sequence of GPR119 in rodents differs from that in humans, making it difficult to translate in vivo models in mice or rats into human clinical trials [37]. To enhance its efficacy, a consideration has been made towards combining it with DPP-4 inhibitors, which inhibit the hydrolysis of GLP-1 and maximize the effect of the GPR119 agonist-induced secretion of GLP-1 [37]. HBK001 is a GPR119/DPP-4 dual-targeting drug with better effects than single-targeting therapy. Li et al. [41] optimized the benzene ring structure of HBK001 and synthesized the hydrochloride form of HBK001 (HBK001 hydrochloride) with an oxadiazole side chain structure, which improved the DPP-4 inhibitory effect and moderate agonism of GPR119, showing improved bioavailability and hypoglycemic effects in vivo, but moderate inhibition of hERG channels, probably due to its high lipophilicity.

GPR120, another class A GPCR [32], is activated by long-chain fatty acids and widely expressed in intestinal endocrine L, K, and I cells, pancreatic β-, δ-, and α-cells, adipocytes, and macrophages [31,42]. Two splice variants of GPR120 exist: the short isomer called GPR120S, and the long isomer called GPR120L. The main difference between these two isomers is the insertion of 16 amino acids in the third intracellular loop of GPR120L, which confers different signaling properties in that while GPR120S couples to both the Gαq/11 and β-arrestin pathways, GPR120L only couples to the β-arrestin pathway [43] (Figure 2). The activation of GPR120 is related to the secretion of somatostatin and ghrelin (a type of hunger hormone) and inhibits the secretion of ghrelin, thereby inhibiting the secretion of somatostatin via pancreatic δ-cells and gastric D cells, which eliminates the inhibitory effect of somatostatin on insulin secretion. GPR120 also promotes the secretion of incretin [31]. Currently, no GPR120 agonist has entered clinical trials.

However, in general, due to the lipophilic nature of the ligands of GPCRs and the biased nature of signal transduction, as well as the unresolved crystal structures of GPR120 and GPR119, the development of single-targeted drugs with high efficacy and safety requires much research.

## 4. Glucose Metabolism Pathway-Based Targets and Therapeutic Medicines

### 4.1. Glucose Kinase (GK)

GK, also known as hexokinase IV, is a glycolytic enzyme mainly involved in the first step reaction of glucose metabolism, phosphorylating glucose to form glucose-6-phosphate [44]. GK is primarily found in pancreatic β-cells and hepatocytes and expressed in intestinal endocrine cells, neurons, pancreatic α- and δ- cells, and anterior pituitary cells [45]. GK has a higher Km value compared with the other three subtypes of the hexokinase family (I–III) and its activity is not inhibited by its product glucose-6-phosphate [46]. The three-dimensional structure of GK can be divided into three parts, namely the large domain, small domain, and linker domain, of which the linker domain is the main active region of GK, where the glucose binding site is located [46]. Depending on the substrate binding state, GK exists in three conformations, namely closed, open, and super-open; open and closed correspond to the substrate acceptance state and phosphorylation reaction state, respectively, and super-open denotes the inactive state [46]. GK exerts different regulatory mechanisms in β-cells and hepatocytes to maintain glucose homeostasis. In β-cells, as blood glucose levels rise, GK phosphorylates glucose and enters the glycolytic, tricarboxylic acid cycle (TCA) pathway, producing large amounts of ATP, leading to the inhibition of ATP-sensitive potassium channel (K_ATP_) activity on the β-cell surface, allowing for an increased inward flow of Ca^2+^, and ultimately GSIS secretion [46] (Figure 3). In the liver, GK regulatory protein (GKRP) is a competitive inhibitor of glucose, forming an inactive compound with GK during hypoglycemia, which is then replaced with glucose as blood glucose levels increase. The GK–GKRP complex subsequently dissociates, and GK activity increases to regulate blood glucose levels by synthesizing hepatic glycogen after phosphorylating glucose [45,46] (Figure 4).

Although GK activity is generally low in patients with T2DM, GK agonists (GKAs) can be developed to assist in their therapy. GKAs also bind to the linkage domain of GK, but at a distance from the glucose-binding site, so that the binding of GKAs promotes the shift to the active conformation of GK without affecting the binding of glucose and can accelerate the phosphorylation reaction [46]. Available GKAs can be classified as hepatic-selective GKAs, hepatopancreatic dual-acting partial GKAs, and hepatopancreatic dual-acting complete GKAs [47]. Most GKAs display hypoglycemic capacity, but they also suffer from dyslipidemia, fatty liver induction, and a poor long-term efficacy. This lack of sustained efficacy may be due to the toxic effect of GKAs on β-cells [48], which leads to β-cell dysfunction and further triggers hyperglycemia [49]. GK activity in Gck^+/−^db/db mice is expressed in β-cells at approximately half the level of wild-type mice, while its expression in the liver is unchanged. This phenotype is very similar to that of β-cell GK gene-specific knockout mice, which can be used as a model to investigate whether the long-term efficacy of GKAs can be improved by inhibiting some GK activity to ameliorate β-cell failure caused by excessive glycolysis [48,49]. Previous studies have shown that Gck^+/−^db/db mice reduce the expression of metabolic stress-related genes, thereby increasing the expression of β-cell function and maturation-related transcription factors, reducing mitochondrial damage, and improving metabolic patterns [48]. Therefore, the specific knockdown of GK genes on β-cells may improve the poor efficacy of GKAs. Dyslipidemia and fatty liver production may be due to the overactivation of GKs in hepatocytes by GKAs, leading to the excessive accumulation of G-6-P and elevated levels of acetyl coenzyme A, ultimately increasing the influx of fatty acids and triglycerides, as well as hepatic *de novo* lipogenesis [50].

Dorzagliatin, the first GKA approved for marketing by the State Drug Administration of China, improves the glucose sensitivity of β-cells and repairs early insulin release defects, fundamentally addressing the failure to properly sense blood glucose changes due to the deterioration of β-cells caused by GK damage [51,52]. The results of the phase III clinical trial of dorzagliatin demonstrated that it is safe and well tolerated as a monotherapy drug for the treatment of patients with T2DM. Patients’ blood glucose levels improved significantly starting at week 4 of treatment, and these improvements were maintained during a 24-week double-blind treatment period. This was followed with a 28-week open-label trial, with efficacy maintained until week 52 [47]. When dorzagliatin was used as an add-on therapy to metformin, HbA1c levels were reduced by 0.66%, and common deficits in patients at different stages of the disease were addressed [53].

### 4.2. Protein Kinase B (AKT/PKB)

AKT, also known as PKB, is a Ser/Thr kinase that contains three structural domains, which are as follows: a central kinase structural domain with specificity for substrate protein Ser/Thr residues, a pleckstrin homology structural domain (PH) that mediates membrane recruitment, and a carboxy-terminal regulatory structural domain [54,55]. AKTs can be classified into three subtypes, AKT1 (T308/S473), AKT2 (T309/S474), and AKT3 (T305/S472), depending on the activated site [56]. AKT1 is mainly involved in cell proliferation and apoptosis and can regulate body size; AKT2 is mainly involved in maintaining glucose homeostasis; and AKT3 mainly plays a role in brain development [55,57]. The activation of AKTs is mediated by 3′-phosphatidylinositol-dependent protein kinase 1 (PDK1) and rapamycin complex 2 (mTORC2) [58]. Various growth factors, cytokines, and hormones first activate receptor tyrosine kinase, while PI3K, the upstream molecule of AKTs, receives the signal that catalyzes the conversion of phosphatidylinositol 4,5-bisphosphate (PIP2) into phosphatidylinositol 3,4,5-trisphosphate (PIP3), which binds to and activates PDK1, thereby translocating it to the cell membrane. When PDK1 translocates to the membrane, PIP3 then binds to the PH structural domain of AKT, causing AKT to translocate to the membrane and a conformational change in AKT, thereby making the Thr site of AKT more susceptible to phosphorylation by PDK1 [56,59]. Subsequently, mTORC2 phosphorylates the Ser site of AKT, thereby activating it [58].

AKT activation can then act on various effector proteins to mediate various physiological processes, such as glycometabolism, cell proliferation and apoptosis, and the cell cycle (Figure 5). Among the major effector proteins associated with glycometabolism are glycogen synthase kinase 3 (GSK3), AKT substrate of 160 kDa (AS160), and forkhead box O transcription factor (FoxO) [60]. Glucose transporter protein 4 (GLUT4) is an effector of AS160, which is normally present in an inactive state in cytoplasmic vesicles. When AS160 is activated by AKT phosphorylation, GLUT4 translocates to the cell membrane and facilitates glucose entry into the cell [61]. GSK3 inhibits glycogen synthase (GS) under normal physiological conditions, while its phosphorylation by AKT inactivates it, thereby relieving the inhibition of GS, which allows for the glycogen-mediated synthesis of GLUT4-transported glucose and regulation of blood glucose levels [62]. FoxOs are regulators of insulin downstream signaling; when their Ser sites are phosphorylated by AKT, they bind to 14-3-3 proteins, which results in their retention in the cytoplasm, reducing the transcriptional activity of genes related to glucose metabolism in the nucleus and decreasing hepatic gluconeogenesis, thus achieving the effect of regulating blood glucose levels [60].

Therefore, in terms of treatment, AKT activators may be developed for regulating blood glucose. SC79, an AKT activator, is often used to inhibit apoptosis and neuroprotection [63,64]. The use of herbal medicines to activate AKT and thus achieve hypoglycemia is a hot research spot. Wang et al. [65] demonstrated the hypoglycemic and insulin-sensitizing effects of sea cucumber gonad hydrolysates in diabetic rats, which significantly increased the protein expression level of phosphorylated AKT during treatment. Liu et al. [66] demonstrated the antidiabetic activity of cyclic enol ether terpenoids isolated from *Patrinia scabiosaefolia*, which was achieved through upregulating the phosphorylation level of AKT. However, AKT is often over-activated in cancer; therefore, to treat diabetes through the activation of AKT, attention needs to be paid to whether other life processes are affected.

### 4.3. Transforming Growth Factor-β1 Stimulated Clone 22 D4 (TSC22D4)

TSC22D4 is a member of the TSC22 protein family that comprises the subtypes TSC22D1, TSC22D2, and TSC22D3, all of which contain a highly conserved TSC box with a leucine zipper structural motif at their C-terminus, which allows for the formation of homodimers or heterodimers between the subtypes to regulate biological functions. The N-terminal end of TSC22D4 contains two domains termed R1 and R2, and the R2 domain and TSC boxes are connected through a highly conserved disordered region (Figure 6), which contains different post-translational modification sites, mediating the interaction of TSC22D4 with different proteins [67]. Previous studies have shown that TSC22D4 is expressed at high levels in patients with T2DM, and its expression is significantly and negatively correlated with the patients’ glucose consumption. The role of TSC22D4 may be mediated through lipocalin 13 (LCN13). The expression of LCN13 was significantly upregulated through the knock down of TSC22D4, and the phosphorylation level of AKT, a key protein of the insulin pathway, was also upregulated, while the expression of gluconeogenesis-related genes was downregulated. Furthermore, the knockdown of LCN13+TSC22D4 in db/db mice upregulated fasting glucose levels and downregulated glucose tolerance compared with TSC22D4 knockdown mice. This indicated that LCN13 is at least partially involved in the improvement of blood glucose levels brought about by the knockdown of TSC22D4 [68]. Recent studies have further elucidated the glucose regulation mechanism of TSC22D4. TSC22D4 interacts with AKT1, which affects the phosphorylation levels of AKT and its downstream proteins while influencing the expression of genes related to the glucose metabolic pathway. The evidence also suggests that TSC22D4 interacts most strongly with AKT1 during starvation, which is weakened via stimulation with glucose and insulin. Moreover, oxidative stress induced by inhibiting the production of reactive oxygen species from mitochondria can also enhance the TSC22D4–AKTl interaction, resulting in a decrease in the phosphorylation level of AKT-S473 (Figure 7). In addition, by constructing mutants with deletions of certain fragments of TSC22D4, it was found that the disordered region of TSC22D4 is essential for its interaction with AKT1 [67]. Therefore, the development of inhibitors or degraders of TSC22D4 may improve the blood glucose levels in patients with T2DM.

### 4.4. Nemo-like Kinase (NLK)

NLK is an evolutionarily highly conserved, atypical Pro-directed Ser/Thr mitogen-activated protein kinase, which is a direct homolog of the Nemo gene and was first identified in Drosophila [69]. NLK can directly phosphorylate transcription factors or signaling pathway intermediates, acting positively or negatively depending on the regulated physiological process [70]. Previous studies on NLK have focused on resolving its role in the development of cancers, such as liver cancer and breast cancer [71,72]. A recent study, however, showed that NLK may play a key role in the gluconeogenesis regulatory network as a novel negative regulator of hepatic gluconeogenesis and is a potential therapeutic target for T2DM, a role that has not previously been characterized for NLK [73]. NLK primarily represses the gene expression of the key rate-limiting enzymes of the gluconeogenesis process, namely phosphoenolpyruvate carboxykinase (PCK1) and glucose-6-phosphatase catalytic subunit (G6PC). This inhibitory effect is mainly mediated by cAMP response element-binding protein-regulated transcriptional coactivator 2 (CRTC2) and FoxO1. NLK phosphorylates CRTC2 and promotes CRTC2 phosphorylation-driven nuclear export and leads to its proteasomal degradation, thereby inhibiting the expression of G6PC and PCK1. In contrast, FoxO1 is an autotranscription factor of its own gene, and its phosphorylation by NLK drives nuclear export to block the transcriptional activity of its own gene and gluconeogenetic genes; however, FoxO1 is not subjected to proteasomal degradation (Figure 8). The authors also found a significant downregulation of NLK expression levels in animal models of T2DM and the same gene expression profile in patients with T2DM, suggesting that reduced NLK is likely to contribute to the pathogenesis of T2DM. Therefore, NLK activators are a potential treatment strategy against T2DM [73].

## 5. Insulin-Based Targets and Therapeutic Medicines

### 5.1. Fibroblast Growth Factor 21 (FGF21)

The fibroblast growth factor (FGF) superfamily includes 22 family members, which are divided into seven subfamilies, most of which are present in the extracellular matrix and are involved in the regulation of physiological processes by paracrine or autocrine means using acetyl heparin sulfate as a cofactor [74]. In contrast, members of the FGF19 subfamily (FGF19, FGF21, and FGF23) are released from the extracellular matrix into the bloodstream to act in an endocrine manner as they lack the acetyl heparan sulfate-binding domain [75]. Among them, FGF21 has attracted attention owing to its involvement in various physiological pathways, such as glucose and lipid metabolism [76]. FGF21 is mainly secreted by the liver, followed by the adipose tissue, skeletal muscle, and pancreas [77]. Activation of downstream signaling via FGF21 requires the co-involvement of FGF receptor 1 (FGFR1) and co-receptor β-klotho (KLB) [75]. KLB acts as a targeting receptor, first binding to FGF21 and later promoting the binding of FGF21 to FGFR1, and FGFR1 mediates receptor dimerization and participates in intracellular signaling as a catalytic subunit [78]. FGFR1 is commonly expressed in various tissues, whereas KLB is mainly restricted to expression in specific metabolic tissues (e.g., the liver, pancreas, and adipose tissue); thus, KLB confers tissue specificity to the action of FGF21 [75,77].

FGF21 administration promoted weight loss and improved insulin sensitivity in multiple preclinical models [79]. However, human natural FGF21 (hFGF21) is not suitable for clinical use owing to its poor pharmacokinetic properties; its short half-life of 0.5–2 h may be partly due to its small molecular weight (19.5 kDa), which increases its filtration rate through the glomerulus, and partly due to its susceptibility to cleavage by proteases in plasma or its inactivation by aggregation into insoluble proteins under physiological conditions [75,80]. hFGF21 is subject to DPP-4 and fibroblast-activating protein (FAP) cleavage, with DPP-4 responsible for cleavage at the Pro-2 and Pro-4 sites and FAP responsible for cleavage at the Pro-171 site, which is followed by the C-terminus, a KLB-binding site; thus, the cleavage of Pro-171 inhibits its binding to KLB, which results in a loss of activity [75,77]. Therefore, attempts have been made to develop hFGF21 analogs or use gene therapy and stem cell therapy to promote clinical treatment of this target.

Zhu et al. [81] used nuclear magnetic resonance to resolve the solution structure of FGF21 and found that the flexible atypical trilobal conformation in its β10–βl2 region severely affected the folding of the β2–β3 hairpin and the overall protein stability, thus affecting the binding of FGFR1. Therefore, the authors used protein engineering to transfer the natural SS bond on FGF19 to FGF21, constituting the FGF21–FGF19 chimera FGF21SS. In protein stability experiments, FGF21SS exhibited higher thermal stability. Meanwhile, in diabetic mice, FGF21SS had higher biological activity; in inflammatory adipocytes, it inhibited the nuclear factor-κB (NF-κB) signaling pathway, reversed insulin resistance induced by inflammatory factors, and displayed good hypoglycemic, weight loss, and serum insulin-lowering abilities [82]. Queen et al. [83] attempted to utilize gene therapy to treat diabetes, minus the cumbersome need for regular administration of FGF21 analogs. The authors treated insulin-resistant mice with a single low dose of an adenoviral vector encoding FGF21 and showed that their method consistently counteracted insulin resistance without adverse effects, as well as reduced body weight and inflammation. Pan [84] et al. screened out the FGF21 mutant with high affinity to KLB and fused it with the Fc fragment of IgG4 (Fc-FGF21), further prolonging the half-life of the mutant. Furthermore, the GLP-1-Fc-FGF21 double-target fusion protein was constructed. Animal experiments have demonstrated that it can effectively and continuously reduce blood glucose and body weight, improve blood lipids and other indicators, and has better efficacy than the single-target therapy of FGF21 and GLP-1.

Stem cell therapy has been applied to the treatment of T2DM. Xue et al. [85] used adipose-derived mesenchymal stem cells with high expression of GLP-1 and FGF21 genes achieved by lentiviral transduction to treat diabetic mice. The authors revealed that this method significantly reduced blood glucose and body weight, increased insulin sensitivity, and improved the lipid profile.

However, FGF21 treatment tends to inhibit osteoclastogenesis or promote osteoclast differentiation, leading to bone loss. In addition, the hypoglycemic function of many analogs is inactivated in human treatment, probably due to the low abundance of brown adipose tissue in humans compared to that in rodents, which is an important target organ for FGF21. Furthermore, the duration of existing clinical trials of FGF21-based therapies is short, whereas T2DM is a chronic metabolic disease; therefore, longer-term treatment is needed to assess the associated efficacy and safety issues [75].

### 5.2. Protein Tyrosine Phosphatase 1B (PTP1B)

Tyrosine phosphorylation of proteins is one of the mechanisms that regulate cellular function, a reversible process mediated by protein tyrosine phosphatases (PTPs, responsible for dephosphorylation) and protein tyrosine kinases (responsible for phosphorylation). PTP1B belongs to the intracellular PTPs expressed in various cells and tissues, and its encoding gene, *PTPN1*, is located in a region associated with insulin resistance and obesity; hence, it is involved in the regulation of the insulin metabolic pathways [86]. PTP1B consists of three structural domains, which are as follows: the N-terminal catalytic domain, the regulatory structural domain, and the C-terminal domain that targets the endoplasmic reticulum membrane [87]. The N-terminal catalytic domain active site contains Arg, which can generate a positive charge, and Cys, which has nucleophilic activity, both of which are essential for the catalytic action of PTP1B. Arg can stabilize the anion moiety through its charged activity, thereby regulating the nucleophilic activity of Cys and ultimately the enzymatic activity of PTP1B [88]. Further, the phosphorylation of Tyr residues on the catalytic structural domain of PTP1B is one of the mechanisms regulating its activity, which is active when its Tyr 66 residue is phosphorylated; meanwhile, it is inactivated when its Ser 50 residue is phosphorylated [89]. Under insulin signaling, the insulin receptor (IR) is activated via autophosphorylation, which can later phosphorylate the Tyr residue of PTP1B, thereby activating PTP1B, and the activated PTP1B dephosphorylates IR and insulin receptor substrate-1 (IRS-1), which leads to the inactivation of the downstream pathway of PI3K-AKT and prevents the translocation of GLUT4, with a negative insulin metabolic pathway regulatory effect [90] (Figure 5). Therefore, the inhibition of PTP1B could be a strategy for the treatment of diabetes.

However, since PTP1B was identified as a target, the development of its inhibitors has been hampered. The active site of PTP1B is highly positively charged; hence, an effective inhibitor should either be anionically charged or strongly polarized, whereas these properties of the inhibitor weaken its membrane permeability and limit oral bioavailability [90]. Furthermore, the active sites of different proteins in the PTP family are highly conserved and there is a great similarity in the protein sequences and structural features between them, which hinders the development of specific inhibitors for specific members, among which T-cell protein tyrosine phosphatase is the closest homolog to PTP1B [88,90]. However, the allosteric sites among PTPs have relatively unique structures; they are not as highly conserved and highly polarized as the active sites, and when targeting PTP1B inhibitors to the allosteric sites, their druggability has been assessed to be much higher than that when targeting the active sites; however, better inhibition and selectivity are observed if both sites are targeted simultaneously [88,91]. In addition to the allosteric sites, another site specific for PTP1B has been characterized termed the second aryl phosphate-binding site (also called the B site or secondary site), an important non-conserved site for regulating substrate specificity, and targeting this site can also improve the specificity of the inhibitor [90]. Moreover, antisense oligonucleotide (ASO) inhibitors selectively bind specific sequences of PTP1B mRNA and reduce the expression of PTP1B, improving the poor membrane permeability [89]. Although improvement strategies continue to be proposed, the only PTP1B inhibitors that have entered clinical trials to date are ertiprotafib (a non-competitive pleiotropic inhibitor), ISIS-113715 (an ASO inhibitor), trodusquemine (an allosteric inhibitor), and JTT-551 (a hybrid inhibitor); however, all the trials were eventually discontinued due to the emergence of adverse side effects and low specificity [92,93].

Currently, researchers are paying close attention to natural products. Natural products isolated from natural plants have good biocompatibility, low side effects, synergistic effects on a wide range of diseases, and are of low cost with great medicinal value [92]. Many natural products, including phenols, terpenoids, flavonoids, and alkaloids, have been identified as potent PTP1B inhibitors [94]. Casertano et al. [95] demonstrated that avarone, a sesquiterpene quinone extracted from the marine sponge *Dysidea avara*, can improve both insulin sensitivity and mitochondrial activity and is a tight-binding inhibitor of aldose reductase, which prevents the occurrence of diabetic retinal complications. Ali et al. [96] demonstrated that ursonic acid extracted from *Artemisia montana* also inhibits the expression of PTP1B, which activates GLUT4 in the PI3K/AKT signaling pathway and increases peripheral glucose uptake.

The development of PTP1B inhibitors with high specificity, low side effects, and good inhibitory properties warrants further research.

## 6. Other Targets and Therapeutic Medicines

### 6.1. Sodium-Glucose Cotransporter Protein-2 (SGLT-2)

SGLTs belong to the mammalian solute carrier family SLC5, a class of integral membrane proteins that utilize the electrochemical potential of sodium ions to drive the transport of glucose, anions, vitamins, and short-chain fatty acids against the concentration gradient, with six isoforms of which SGLT-2 plays a major role in the kidney [97]. SGLT-2 is expressed mainly in the S1 and S2 segments of the renal proximal tubule, a cotransport protein with high transport capacity and low affinity that reabsorbs 90% of glucose from the glomerular filtrate back into the blood [98]. SGLT-2 is in an overexpressed state in patients with T2DM, which enhances glucose uptake through the kidneys and increases blood glucose levels. Therefore, it is possible to lower blood glucose by inhibiting SGLT-2 in patients to block the renal tubular reabsorption of glucose from primary urine and increase urinary glucose excretion [98]. Moreover, the regulatory mechanism of SGLT-2 inhibitors is insulin-independent; hence, there is generally no risk of hypoglycemia, and it protects β-cells from glucose toxicity [99,100].

Metformin is the drug of choice in the clinical management of diabetes; however, due to individual variability, it is sometimes not the first choice and needs to be complemented with other drugs, and among these drugs, SGLT-2 inhibitors have more effects that can also prevent diabetic complications. SGLT-2 inhibitors have a lower all-cause mortality rate compared with DPP-4 inhibitors and GLP-1R agonists, especially in terms of cardiovascular mortality and hospitalization for heart failure [101,102]. Furthermore, SGLT-2 inhibitors significantly reduce the risk of renal disease [103]. Part of the reason may be the improvement of patients’ high serum uric acid, which is highly susceptible to low-grade inflammation, which in turn is a key driver of vascular complications [104,105]. Another reason may be the sodium-loving effect of SGLT-2 inhibitors, which results in a lower blood pressure in patients and thus promotes cardiovascular and renal protection [106]. However, the unique hypoglycemic mechanism of SGLT-2 inhibitors provides favorable conditions for microbial colonization in the genitourinary tract, which is highly susceptible to genitourinary tract infections and has a higher incidence in women [107]. SGLT-2 inhibitors also increase free fatty acid oxidation and glucagon release, as well as decrease insulin secretion, which may lead to rare and serious complications, diabetic ketoacidosis, and adverse effects, such as fractures, hypovolemia, amputations, and other adverse effects [108,109,110,111].

Currently, the main SGLT-2 inhibitors in clinical use include dapagliflozin, canagliflozin, empagliflozin, and ertugliflozin, which show heterogeneity in specific regulatory mechanisms and adverse effects due to the different structural formulations of these drugs. For example, empagliflozin has the most additional benefits among them in addition to its hypoglycemic effect and is the first hypoglycemic drug in the world to reduce the risk of cardiovascular death in patients with T2DM. It is now included in the World Health Organization Essential Medicines List, increasing the accessibility and affordability of this drug in the future [112,113]. In addition, the increased risk of fracture has been more often considered to be caused by canagliflozin [114].

Henagliflozin was the first original SGLT-2 inhibitor in China, which has undergone a 10-year long research run and was approved for marketing at the end of 2021. Henagliflozin is structurally optimized, resulting in better chemical stability, drug solubility, and high receptor selectivity, with the SGLT-2/SGLT-1 receptor selectivity ratio up to 1823.53 [115,116]. The phase III clinical study showed that henagliflozin has a good hypoglycemic effect on both fasting and postprandial glucose, and when combined with metformin, it provides long-term efficacy (with this effect being able to persist for 28 weeks following treatment termination) [116,117]. In addition, henagliflozin provided multiple benefits, such as weight loss and blood pressure reduction, and has a good safety and tolerability profile, with an incidence of adverse events comparable to that of the placebo with no additional risk of hypoglycemia and urinary tract infection [116].

### 6.2. Peroxisome Proliferator-Activated Receptor (PPAR)

PPARs are ligand-dependent transcription factors of the nuclear receptor superfamily that exist in three subtypes, namely PPARα, PPARβ/δ, and PPARγ, with a high degree of structural homology, but differ in their functional roles, tissue distribution, and ligand-binding properties [118,119]. PPARα is expressed mainly in the liver, heart, skeletal muscle, BAT, intestine, and kidney, and regulates the function of fatty acids by affecting their transport, esterification, and oxidation [120]. PPARβ/δ is expressed in all organs, but is highly expressed in the skeletal muscle, adipose tissue, and heart, and is mainly involved in the regulation of fatty acid oxidation and blood glucose levels [119,120]. PPARγ has three main allosteric isoforms, PPARγ1, PPARγ2, and PPARγ3. PPARγ1 is expressed in almost all cells, whereas PPARγ2 is mainly restricted to the adipose tissue but can be induced in other tissues on a high-fat diet, and PPARγ3 is predominantly expressed in the macrophages and white adipose tissue, where PPARγ mainly regulates adipocyte differentiation [118], lipid storage, and insulin sensitivity [118,121,122]. PPARs have four major functional domains, which are as follows: the N-terminal non-ligand-dependent transcriptional activation domain (A/B structural domain), where phosphorylation of this region leads to the inhibition of the transcriptional activation function of PPARs; the DNA-binding structural domain, also known as the C structural domain, responsible for binding to the peroxisome proliferator response element (PPRE) on the promoter of PPAR target genes; the D structural domain, which is the docking site for various cofactors; and the large Y-shaped hydrophobic binding pocket formed at the C-terminus, which is the ligand-binding structural domain (E/F structural domain) that confers the ability to bind endogenous or exogenous lipophilic ligands to PPARs [120]. As shown in Figure 9, under physiological conditions, PPARs form heterodimers with the retinoid X receptor (RXR) and bind to the repressor protein in a repressed state. Upon binding of the ligand, the PPAR conformation changes, prompting the dissociation of the corepressors and recruiting of the coactivators, which subsequently regulates the transcription of the target gene by binding to the PPREs on the target gene promoter [123].

PPARγ is the target of the thiazolidinedione (TZD) class of antidiabetic drugs, including the TZD derivatives pioglitazone and rosiglitazone [118]. TZDs act as ligands, and activation of PPARγ upregulates genes related to the glucose metabolism pathway (including GLUT4, IRS-1, IRS-2, and c-Cbl associated protein) and insulin-sensitizing adipokine gene expression (lipocalin) and downregulates the expression of cytokine genes involved in the induction of insulin resistance (tumor necrosis factor-α [TNF-α] and interleukin-6 [IL-6]), thereby regulating blood glucose levels [124,125]. In addition, PPARγ activation promotes the uptake and storage of free fatty acids in adipose tissue and reduces the fat content accumulated in non-adipose tissues, such as the liver, muscle, and pancreas, thereby attenuating metabolic damage from lipotoxicity, promoting glucose utilization, and improving insulin sensitivity [126]. However, TZDs are likely to over-activate PPARγ, which promotes the expression of the Na transporter in the renal collecting ducts, leading to the occurrence of fluid retention, weight gain, and heart failure, which is the reason for the withdrawal of most TZD analogs from their use [124]. Based on this, investigators set out to develop multiple agonists, such as PPARα/γ, PPARα/δ, and PPARα/δ/γ, to moderate the activation of each subtype, thereby alleviating the adverse effects associated with the overactivation of PPARγ mono-subtypes.

Chiglitazar sodium is the first PPAR full agonist independently developed in China and is approved for the treatment of T2DM. It was included in the national health insurance program in January 2023. Owing to the simultaneous moderate activation of three functionally different but overlapping PPAR subtypes, chiglitazar sodium attains the ability to not only selectively alter the expression of a series of genes related to insulin sensitivity, targeting insulin resistance, which is one of the core pathological mechanisms for the occurrence and development of T2DM, but also regulates the symptoms of metabolic syndromes, such as hypertension and dyslipidemia, that often accompany patients with diabetes (https://www.chipscreen.com/products/615.html (accessed on 22 June 2023)). The results of phase III clinical trials of Chiglitazar sodium showed that it significantly reduced HbA1c levels by 1.52%, and its hypoglycemic efficacy was stable for up to 52 weeks, with improvements in fasting and 2 h postprandial glucose compared with that with sitagliptin. It also significantly improved insulin sensitivity, protected islet secretion, and reduced triglyceride levels. In addition, the safety and tolerability were also deemed to be good, and the number and severity of adverse reactions were comparable to those of the placebo and sitagliptin; the incidence of edema was lower and so was the rate of weight gain than those of the TZDs [127,128].

### 6.3. Gut Microbiota

The gut microbiota is a complex ecosystem consisting of microbial communities present in the human gut [129]. When the diversity and abundance of bacteria in the gut microbiota are significantly altered, the gut microbiota becomes dysregulated, which is the driving factor of chronic low-grade systemic inflammation, leading to chronic non-infectious diseases, such as obesity and T2DM [130]. The bacteria enriched in patients with T2DM are mainly conditionally pathogenic, such as *Bacteroides caccae*, *Escherichia coli*, and *Eggerthella lenta*, and they reduce abundance of butyrate-producing bacteria, including *Eubacterium rectale*, *Faecalibacterium prausnitzii*, and *Roseburia intestinalis* [131]. Although the specific mechanisms through which gut microbiota affect the development of T2DM are complex, some of the clearer mechanisms involve the short-chain fatty acid (SCFA), bile acid (BA), branched-chain amino acid (BCAA), endotoxin-intestinal barrier, imidazole propionate (ImP), and aryl hydrocarbon receptor (AhR) theories [132].

SCFAs are organic fatty acids containing two to six carbon atoms, which are metabolites produced through anaerobic bacterial fermentation of non-digestible dietary fibers for their own energy needs, with acetate, propionate, and butyrate being the main components [133]. Among them, acetate has the ability to cross the blood–brain barrier, where it activates acetyl coenzyme A carboxylase and upregulates neuropeptide expression, thereby activating hypothalamic neurons and suppressing appetite, reducing the inflammatory response associated with weight gain [134]. Propionate is a substrate for intestinal gluconeogenesis, which enters the TCA cycle and is converted into oxaloacetate, a gluconeogenic precursor [134]. Propionate protects islets by inhibiting apoptosis induced by inflammatory factors and free fatty acids [135]. Butyrate can bind to GPR41 and GPR43 to protect β-cells from oxidative stress, inhibit the deacetylation of histones, downregulate the expression of genes related to gluconeogenesis in the liver, and bind to GLP-1R to promote insulin secretion [136].

BAs are small molecules synthesized in hepatocytes from cholesterol, and their main function constitutes the digestion and absorption of lipids and fat-soluble vitamins [137,138]. The main gut microbiota are involved in the synthesis, modification, and signal transduction of BAs. These microbiota have the ability to produce bile salt hydrolases, enzymes that convert primary BAs into secondary BAs. This process is crucial for the biological functions of BAs [139]. Primary BAs can bind to and activate the farnesoid X receptor (FXR), which improves glucose tolerance and increases insulin sensitivity [134,138]. Secondary BAs bind to Takeda-G-protein receptor-5 (TGR-5) and induce GLP-1 secretion, thereby promoting insulin secretion [129].

BCAAs are mainly valine, isoleucine, and leucine [138]. Chronic increases in plasma BCAA concentrations are predictive markers of insulin resistance and T2DM risk [140]. Increased BCAA leads to mitochondrial dysfunction and sustained activation of the mTOR signaling pathway, which increases Ser phosphorylation of IRS-1 and inhibits the GLUT4 transport of glucose, ultimately leading to insulin resistance [141].

The endotoxin-intestinal barrier theory has been thought to underlie the systemic chronic low-grade inflammation in patients with T2DM [131]. Disturbances in the gut microbiota of patients with T2DM are accompanied with an increase in the proportion of Gram-negative bacteria, which in turn promotes an increase in lipopolysaccharide (LPS) secretion. LPS is recognized by innate toll-like receptors, which later activate the nuclear factor kappa-B (NF-κB) signaling pathway, promoting the release of pro-inflammatory factors, thereby leading to inflammation. This in turn triggers β-cell destruction, insulin resistance, and increased intestinal wall permeability [130,134]. When the intestinal barrier is damaged, intestinal bacteria and their products (LPS) are highly likely to enter the circulation through the damaged barrier, further triggering endotoxemia and systemic chronic low-grade inflammation, thus forming a vicious cycle that leads to further deterioration of insulin resistance [131,139].

ImP is a histidine-derived metabolite elevated in prediabetic and T2DM patients [132], and has been shown to impair insulin signaling by activating mTORC1 and increasing Ser phosphorylation of IRS-1 [142]. Furthermore, it impairs the effects of metformin by inhibiting AMPK activity [143]. Tryptophan is an essential amino acid that is metabolized by intestinal microorganisms to form indoles and their derivatives, some of which serve as AhR ligands. When the gut microbiota is unbalanced, the ability to convert tryptophan into AhR ligands is reduced, and inhibition of the AhR pathway will lead to the decreased production of GLP-1 and interleukin-22, resulting in an increased intestinal permeability and LPS translocation [138].

Probiotic or prebiotic supplementation and fecal microbiota transplantation (FMT) have been proposed as therapeutic approaches based on gut microbiota. Probiotics are active microorganisms located in the gut and exhibit a positive effect on the host [131]. Prebiotics are food components that work through probiotics that beneficially affect the host by promoting the growth of beneficial bacteria [134]. Mixtures of probiotics and prebiotics or combinations of both with antidiabetic drugs have a superior efficacy than probiotics or prebiotics alone [144,145]. FMT is a therapeutic approach that transplants fecal microbiota from a healthy donor into the recipient’s gut, which not only reverses the recipient’s gut microbial dysbiosis, but also rebuilds the gut microbiota ecosystem, and is considered to have better potential than probiotic or prebiotic supplementation [134,146]. However, there are safety concerns; one case of severe bacteremia and one death have been reported. The FDA has proposed more stringent donor screening and testing for multi-drug-resistant organisms (MDROs) [147]. The American Gastroenterological Association has also suggested that patients treated with FMT should be followed up ten years to assess the long-term safety and efficacy of this treatment [131].

### 6.4. microRNA (miRNA)

miRNAs are a class of non-coding, single-stranded RNA molecules encoded by endogenous genes with a length of approximately 22 nucleotides, which play a regulatory role on target mRNAs by destabilizing and inhibiting the translation of target mRNAs. A single miRNA can regulate the expression of multiple target mRNAs, and each mRNA can be regulated by multiple miRNAs. Cells can release miRNAs in free form or in complex with extracellular vesicles, which can be taken up through other types of cells, thus mediating cell-to-cell actions [148]. miRNAs can influence insulin signaling by affecting the expression of INSR and IRS-1, the translocation of GLUT4, and the activity of PI3K; furthermore, they influence insulin secretion by regulating β-cell metabolic stress, proliferation, and survival, as well as regulating GSIS and improving insulin sensitivity [148,149,150,151]. Therefore, miRNAs are potential biomarkers for diabetes prediction [152]. A meta-analysis revealed that miR-29a-3p, miR-221-3p, miR-126-3p, miR-26a-5p, miR-503-5p, miR-100-5p, miR-101-3p, mIR-103a-3p, miR-122-5p, miR-199a-3p, miR-30b-5p, miR-130a-3p, miR-143-3p, miR-145-5p, miR-19a-3p, and miR-311-3p (in order of importance) fulfill the criteria for biomarker selection [153]. However, there is heterogeneity in the expression of miRNAs, and some studies have found sex variability in their use as markers, which may be owing to the following mechanisms: estrogen regulates the transcription and processing of miRNAs, incomplete X-chromosome inactivation leading to the biallelic expression of miRNAs, and regulation of miRNA expression through epigenetics [154]. To date, no miRNA-based antidiabetic therapies have been approved by the FDA [148].

### 6.5. Glucose-Sensitive Neurons (GSNs)

The hypothalamus is a central part of the nervous system involved in the regulation of energy metabolism, and its dysfunction can lead to the onset of systemic metabolic disorders, such as obesity and diabetes. GSNs are present in the middle and base layers of the hypothalamus, which are characterized by glucose sensing, but their responses to changes in glucose levels are not identical. Glucose-excited (GE) neurons show increased activity at high glucose levels and decreased activity at low glucose levels, whereas glucose-inhibited (GI) neurons show the opposite characteristics [155]. A recent study has shown that GSNs hold promise as new targets for antidiabetic therapy. In this study, FGF4 was administered to T2DM mice through the lateral ventricle, and a single administration produced durable glucose-controlling effects for up to 7 weeks or more. Exploration of this mechanism revealed that GSNs and their highly expressed FGFR1 are the key target cells and preferred receptor subtypes that mediate the regulation of persistent glucose homeostasis by FGF4. Additionally, further knockdown experiments confirmed that GI plays a key role [156].

### 6.6. Carbohydrate Response Element-Binding Protein (ChREBP)

ChREBP is a glucose-responsive transcription factor with two isoforms, ChREBPα and ChREBPβ. Deletion of ChREBP has been shown to prevent glucotoxicity and glucose-mediated β-cell death. However, recent findings have suggested that ChREBP is required for glucose-stimulated β-cell proliferation. Overexpression of ChREBPβ leads to glucose toxicity and its subsequent death of β-cells, while overexpression of ChREBPα enhances glucose-stimulated β-cell proliferation, as it stimulates the Nrf2 antioxidant pathway, thereby preventing oxidative damage [157]. Therefore, the development of ChREBPα selective activators are a potential treatment against diabetes.

### 6.7. Islet Microexons (IsletMICs)

Microexons are DNA sequences that encode proteins, approximately ranging from three to twenty-seven nucleotides long, which can be selectively spliced in neurons, microglia, embryonic stem cells, and cancer cells to produce cell type-specific protein isoforms [158]. A recent study found that IsletMICs can be spliced into pancreatic islet cell mRNAs and play an important role in islet function and glycemic control. IsletMICs are regulated by the RNA-binding protein SRRM3, and both SRRM3 and IsletMICs are induced by elevated glucose levels. If depletion of SRRM3 in human and rat β-cell lines, as well as mouse pancreatic islets, or the use of antisense oligonucleotides to inhibit specific IsletMICs, can result in inappropriate insulin secretion, this suggests that IsletMICs are present at low levels in patients with T2DM; hence, upregulation of IsletMICs levels is a potential therapeutic strategy [159].

## 7. Diet- and Exercise-Based Therapy

Although there is a genetic predisposition to T2DM, the onset and progression of the disease are influenced more by the environment, with an unhealthy diet and lack of exercise being the major factors. Despite continuous attempts to develop new drugs, side effects remain challenging, and even the first-line drug metformin can cause nausea and gastric distension. There is also the issue of compliance with medication, and symptoms may not improve once the medication has been stopped. Therefore, intervention through implementing good lifestyle habits, such as proper diet and exercise, with medication as an adjunctive therapy, will be a lower-cost and healthier treatment option.

Diet is a cornerstone in the prevention and treatment of abnormal glucose metabolism and insulin resistance [160]. The ADA released an updated consensus report on diabetes nutrition therapy in 2019 [161]. The report emphasized that there should not be a “one-size-fits-all” dietary pattern for people with diabetes; however, individualized choices should be made as no dietary pattern is completely healthy and can be applied to everyone, but regardless of the pattern, it is generally based on three basic principles, which are as follows: consuming more non-starchy vegetables (such as broccoli, kale, and mushrooms), less sugar and refined grains, and choosing natural foods over highly processed foods. The ketogenic diet (KD) is one of the more researched dietary patterns today, which is a low-calorie, low-carbohydrate, high-fat, and protein-appropriate diet that uses ketone bodies as a source of energy and mimics the fasting state without causing the negative consequences of starvation [162]. A meta-analysis revealed that the ketogenic diet was effective in improving the glycemic parameters, body weight, and lipid profile of patients [163]. Randomized controlled trials have also shown that a ketogenic diet reduces HbA1c and triglyceride levels in patients [164]. Furthermore, it has been observed that diabetes leads to dysregulation in the anti-inflammatory and oxidative stress pathways. However, the implementation of a KD has shown potential in mitigating these detrimental effects. The KD exerts its anti-inflammatory effects through various mechanisms, such as adenosine, ketone bodies, the mTOR pathways, PPARγ, NLRP3 inflammasome, and gut microbiota. Additionally, the KD exhibits antioxidant properties by improving mitochondrial dynamics, regulating miRNA expression associated with oxidative stress, and enhancing the pentose phosphate pathway and related antioxidant defense systems [165,166,167,168]. However, it should be noted that a specific ketogenic diet plan may lead to metabolic ketoacidosis if it is not well designed [169]. In addition, the Mediterranean diet, low-sugar diet, and medium-carbohydrate diet have shown better control of HbA1c and fasting blood glucose levels in patients [170]. Some micronutrients, such as ω-3, polyphenols, and vitamin D, have also demonstrated a correlation with T2DM. Vitamin D reduces the risk of the disease in prediabetic patients [171], while polyphenols have anti-inflammatory and antioxidant properties, reduce β-cell apoptosis, inhibit α-glucosidase, modulate the intestinal microbiota, and improve adipose tissue metabolism [172]. While ω-3 appears to have different efficacy, a previous large-scale prospective study showed that ω-3 reduces the risk of T2DM [173]; however, a recent mendelian randomized study showed that ω-3 can act through two different clusters of genetic variants, with one set of variants ameliorating insulin resistance, and the other set contributing to β-cell dysfunction and increasing T2DM risk [174]. It is believed that potentially harmful diets should be completely eliminated and beneficial diets should be completely followed; however, diets encompass numerous complexities. Excessive fiber intake can lead to belly bloating and diarrhea; abstainers have a higher prevalence of the disease than moderate drinkers; and indiscriminate combinations and large stacks of macronutrients are counterproductive [161].

Exercise is also a cornerstone for maintaining glucose homeostasis in patients with T2DM, as it increases their overall energy consumption and specifically promotes the uptake of peripheral glucose through the skeletal muscle in an insulin-independent manner [175], which attenuates endoplasmic reticulum stress caused by the overproduction of insulin by β-cells due to overnutrition. Moreover, reducing glucose entry into β-cells for aerobic metabolism directly reduces ROS production and attenuates oxidative stress [176]. A recent large cohort study showed that appropriate exercise increases cardiorespiratory fitness, and that a higher cardiorespiratory fitness reduces T2DM risk [177]. The American College of Sports Medicine updated the Consensus Statement on exercise in patients with T2DM in 2022 [178]. Improvements in systemic insulin sensitivity in patients can last from 2 to 72 h after exercise and the reduction in blood glucose is closely related to the duration and intensity of exercise. Regular aerobic exercise improves glycemic management, reduces time to daily hyperglycemia, and lowers HbA1c levels in adult patients with T2DM, as demonstrated from a meta-analysis on Baduanjin [179]. For elderly patients, resistance training and flexibility training are more suitable for their physical condition. Breaking up sedentary behavior by doing “little and often” daily physical activity can moderately lower postprandial blood glucose levels, especially in patients with insulin resistance and high BMIs. Exercise safety should also be emphasized. In cases where blood glucose levels exceed 250 mg/dL, exercise should be prohibited. Additionally, patients who are on insulin or insulin secretagogues should have fast-acting carbohydrates, such as candy, readily available to prevent hypoglycemia [178].

In conclusion, a healthy diet and regular exercise can reduce the risk of developing T2DM and benefit patients with T1DM [180,181].

## 8. Prospects

Diabetes is a public health issue of global concern as a chronic multifaceted disease that remains incurable and is accompanied with many complications, such as cardiovascular disease and emotional-cognitive dysfunction. In addition, diabetes is expensive to treat, as patients can only delay the end of the disease by constantly taking medications. Concurrently, owing to the large population base, the prevalence of diabetes in China will only increase as the population aging process continues to accelerate. Based on this, efforts have been made to explore the pathogenesis and search for various potential targets to provide opportunities for drug development. Currently, in addition to the mainstream drug metformin, the drugs that occupy a large share of the market comprise mainly GLP-1 agonists, such as semaglutide, which has weight loss, hypoglycemic, and cardiovascular protective effects, and as a long-acting drug, it can be injected once a week, eliminating the tediousness of daily dosing. In addition, there are others, such as SGLT-2 inhibitors, DPP-4 inhibitors, GK agonists, and GLP-1/GIP receptor dual agonists. Stem cell therapy is also making its debut in the field of diabetes. Stem cells can survive for a long time in the body and continuously secrete cytokines, which will help patients to avoid long-term drug injections. However, its operation is complicated and costly, and there are many uncontrollable factors in the process of cell injection, which may pose safety risks. These limitations may affect the universal applicability of this method. Despite the continuously proposed treatment strategies, China still has the highest number of patients with diabetes in the world. Ultimately, individual differences, diversity of pathogenesis, and drug resistance have a significant influence. Different patients may have different underlying diseases, and drugs targeting a certain target may not be suitable for everyone. For instance, metformin is not suitable for patients with heart failure or renal dysfunction. Therefore, an in-depth study of the pathogenesis of diabetes is a long and challenging task, requiring prolonged persistence. Understanding of the pathogenesis, especially the key molecules, proteins, and pathways remains essential. Moreover, the regulatory mechanisms in the body are complex, and cytopathic lesions caused by diabetes may compromise other physiological processes, leading to complications of varying severity.

Therefore, future studies must investigate the changes in the omics of lesioned cells and the specific metabolic pathways they affect. The early prevention and control of diabetes along with reducing the risk of diabetes, by maintaining healthy lifestyle habits and decreasing intake of high-sugar and high-fat diets, should also be a focus. In conclusion, the treatment and prevention of diabetes requires a concerted effort by all parties involved and not just scientists/physicians.

## Figures and Tables

**Figure 1 ijms-24-13381-f001:**
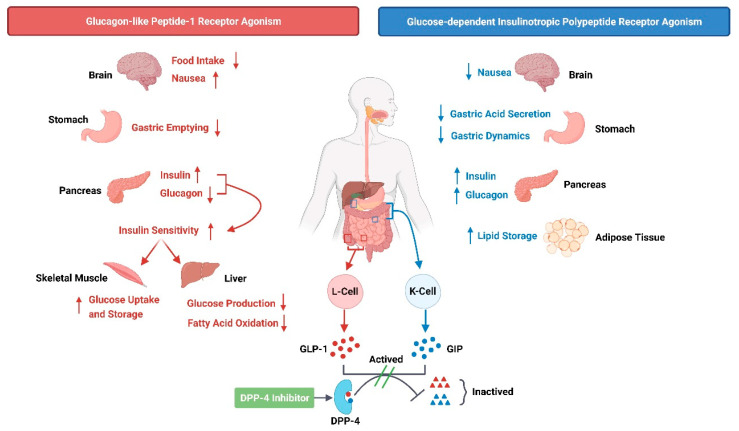
Mechanisms underlying GLP-1, GIP, and DPP-4 glucose regulation. Abbreviations: GLP-1, glucagon-like peptide-1; GIP, glucose-dependent insulinotropic polypeptide; and DDP-4, dipeptidyl peptidase-4. Red arrows: mechanism of GLP-1; Blue arrows: mechanism of GIP; Gray arrows: mechanism of DPP-4.

**Figure 2 ijms-24-13381-f002:**
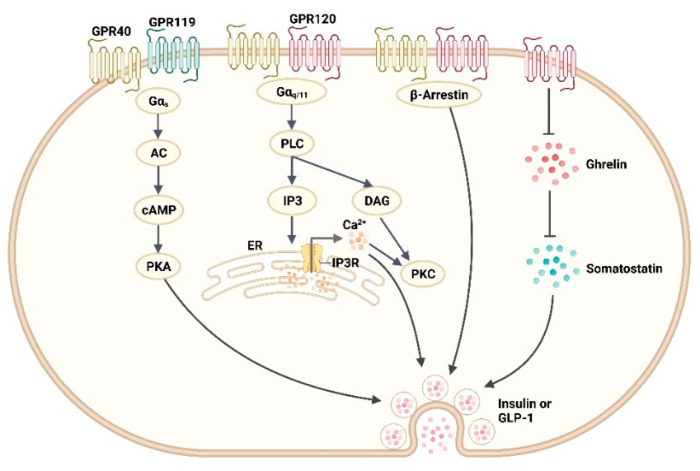
GPR40/GPR120/GPR119 blood glucose regulation mechanism. Abbreviations: AC, adenylyl cyclase; cAMP, cyclic adenosine monophosphate; PKA, protein kinase A; PLC, phospholipase C; IP3, inositol 1,4,5-triphosphate; PKC, protein kinase C; DAG, diacylglycerol; and ER, endoplasmic reticulum. Tip arrows: activation; Flat arrows: inhibition.

**Figure 3 ijms-24-13381-f003:**
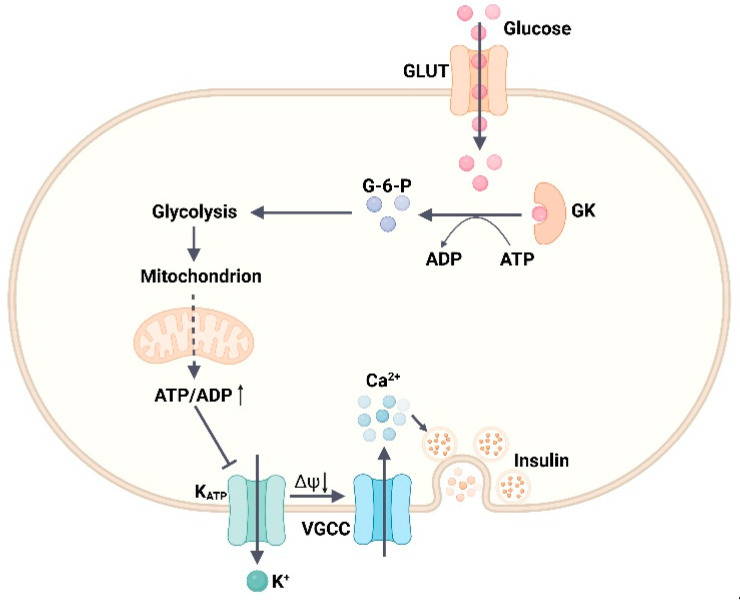
Blood glucose regulation mechanism of glucose kinase (GK) in β-cells. Abbreviations: GLUT, glucose transporter protein; G-6-P, glucose-6-phosphate; ADP, adenosine diphosphate; ATP, adenosine triphosphate; and VGCC, voltage-gated calcium channel. Solid tip arrows: one-step activation; Flat arrows: inhibition; Dashed tip arrows: non-one-step activation.

**Figure 4 ijms-24-13381-f004:**
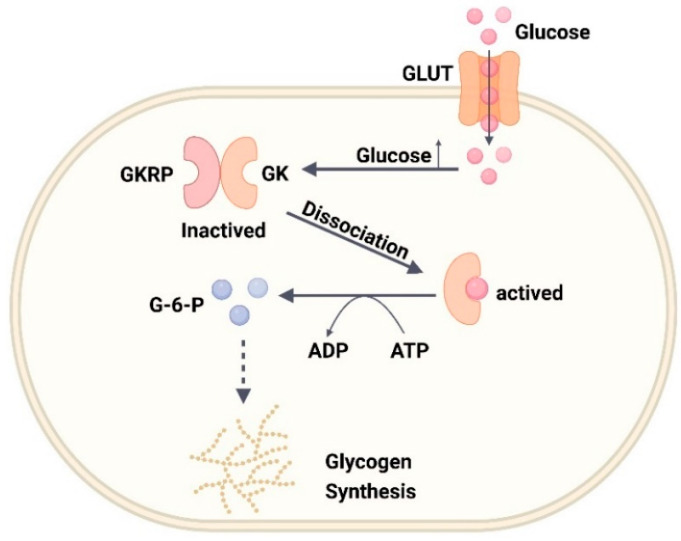
Blood glucose regulation mechanism of GK in the liver. GKRP, GK regulatory protein. Solid tip arrows: one-step activation; Flat arrows: inhibition; Dashed tip arrows: non-one-step activation.

**Figure 5 ijms-24-13381-f005:**
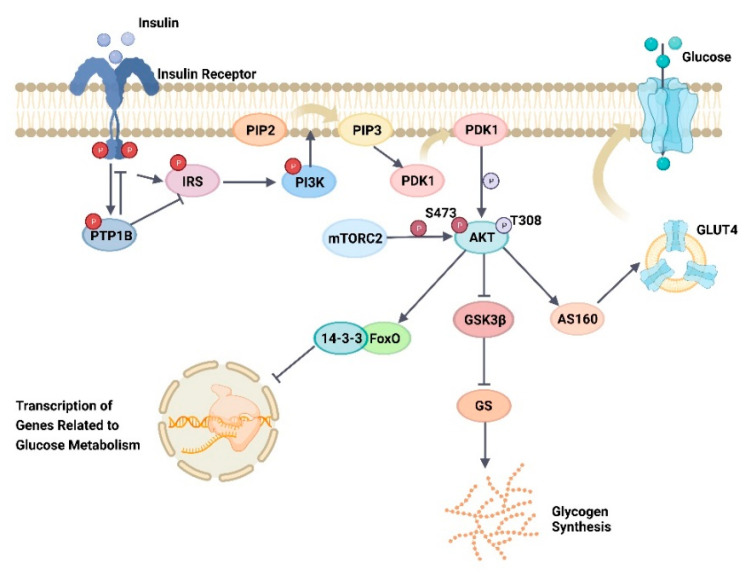
The AKT blood glucose regulation mechanism. Abbreviations: AKT, protein kinase B; PIP2, phosphatidylinositol 4,5-bisphosphate; PIP3, phosphatidylinositol 3,4,5-trisphosphate; PDK1, 3′-phosphatidylinositol-dependent protein kinase 1; PI3K, phosphoinositide 3-kinase; IRS, insulin receptor substrate; PTP1B, protein tyrosine phosphatase 1B; mTORC2, rapamycin complex 2; FoxO, forkhead box O transcription factor; GSK3, glycogen synthase kinase 3; GS, glycogen synthase; AS160, AKT substrate of 160 kDa; and GLUT4, glucose transporter protein 4. Tip arrows: activation; Flat arrows: inhibition.

**Figure 6 ijms-24-13381-f006:**
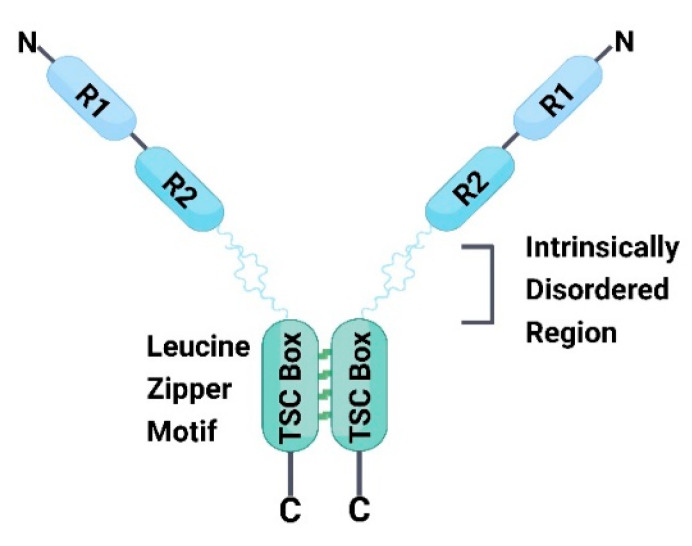
Transforming growth factor-β1 stimulated clone 22 D4 structure diagram.

**Figure 7 ijms-24-13381-f007:**
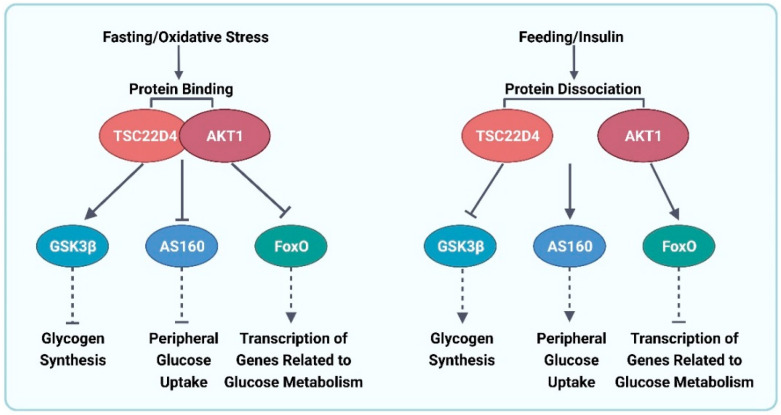
The TSC22D4 glucose regulation mechanism. Solid tip arrows: one-step activation; Solid flat arrows: one-step inhibition; Dashed tip arrows: non-one-step activation; Dashed flat arrows: non-one-step inhibition.

**Figure 8 ijms-24-13381-f008:**
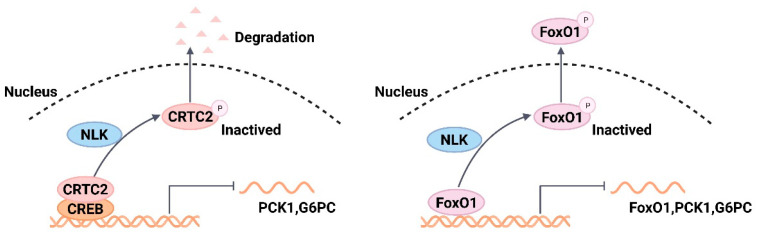
NLK blood glucose regulation mechanism. (**Left**) NLK regulates blood glucose by inhibiting the nuclear output of CRTC2. (**Right**) NLK regulates blood glucose by inhibiting the nuclear output of FoxO1. Abbreviations: NLK, Nemo-like kinase; CRTC2, cAMP response element-binding protein-regulated transcriptional coactivator; CREB, cAMP response element-binding protein; 2PCK1, phosphoenolpyruvate carboxykinase; and G6PC, glucose-6-phosphatase catalytic subunit. Tip arrows: activation; Flat arrows: inhibition.

**Figure 9 ijms-24-13381-f009:**
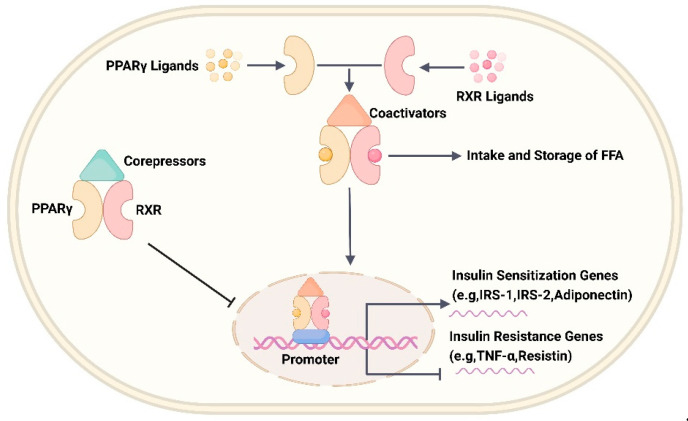
Schematic representation of PPARγ activation. PPAR, peroxisome proliferator-activated receptor; RXR, retinoid X receptor. Tip arrows: activation; Flat arrows: inhibition.

## Data Availability

Data available on request from the authors.

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
