# Peer review of "Advances in Research on Type 2 Diabetes Mellitus Targets and Therapeutic Agents"

_ijms, 2023, doi:10.3390/ijms241713381_

Round 1
Reviewer 1 Report
The manuscript entitled „Advances in Research on Type 2 Diabetes Mellitus Targets and Therapeutic Agents” presents interesting issue, but some problems should be corrected.
Major:
There are really important problems associated with the prepared manuscript, including:
(1) It is not clear what was the aim of the study, as Authors did not formulate it. Based on the title, it may be supposed that Authors intended to present novel knowledge which we have in 2023 while compared with the previous studies/ previous reports. However, there is a very serious problem, as Authors do not present what is new in our knowledge, but reproduce well known information (which were already presented in a number of papers), but novel knowledge is not even mentioned.
(2) From 146 references only 1 (one!) was published in 2023, while 76 of them was published until 2020. Taking into account how quick the knowledge is broadened, the referred studies are old (if Authors intended to present ‘advances’ in the studied area).
(3) The serious flaw of the presented manuscript is associated with the fact, that it presents a highly subjective review, not a systematic review. While the systematic review has a key role for broadening knowledge, the other reviews don’t have such role.
(4) This problem is noted while we compare the ‘advances’ described by Authors with really novel drug targets for type 2 diabetes, which are presented by other Authors, e.g. In the study like https://www.ncbi.nlm.nih.gov/pmc/articles/PMC8869656/ - even if published in 2020, Authors of this study present a number of novel therapeutic options – emerging targets, which Authors of the presented study do not even mention.
(5) Taking into account, that the Materials and methods section is not presented (it should be added), without any specific information, it is hard to understand which areas were included, and consequently which studies were included into review and why. Authors did not present any key words, which were used during literature search, inclusion and exclusion criteria of references, information about the procedure of literature search conducted by them, number of chosen references, as well as information if some of them were excluded from the review and on the basis of which criteria. As a number of recent publications that are related to the issue were not included, it is a serious problem.
(6) Authors do not present the current and comprehensive knowledge associated with the issue. It is associated with the fact that they did not include some important issues, while other were included even if they are not so crucial (or not novel).
(7) Last but not least, for the review articles proper referring the source of information is crucial, while Authors do not do it properly, suggesting that their review just reproduced observations by other authors of review articles. It may be noted from the first sentence, as Authors wrote that “In 2018 the American Diabetes Association classified diabetes mellitus into four major categories […] [1].” – as they mentioned American Diabetes Association, it may have been supposed that they referred American Diabetes Association. But they did not. They referred the review article “Type 2 Diabetes Mellitus: A Review of Multi-Target Drugs” (https://www.mdpi.com/1420-3049/25/8/1987) which presented state of knowledge for 2020. In this study the position by American Diabetes Association from 2018 was referred, but since this moment the novel position of American Diabetes Association was published in 2022 (https://pubmed.ncbi.nlm.nih.gov/34964875/). Authors not only refer improper reference, but also present false information about 2018.
Author Response
August 11, 2023
Prof. Dr. Maurizio Battino
Editor-in-Chief
Mr. Bryan Li,
Assistant Editor
International Journal of Molecular Sciences
Dear Editor,
I wish to re-submit the manuscript titled “Advances in Research on Type 2 Diabetes Mellitus Targets and Therapeutic Agents.” The manuscript ID is ijms-2532957.
Please accept our sincere appreciation for the valuable suggestions and insightful perspectives provided by you and the reviewers. These astute recommendations have greatly enhanced the manuscript.
Attached is the revised version of our manuscript. In the following pages are our point-by-point responses to each of the comments of the reviewers. Revisions in the text are highlighted by the utilization of the color red. We hope that the revisions in the manuscript and our accompanying responses would be sufficient to make our manuscript suitable for publication in International Journal of Molecular Sciences.
Thank you for your consideration. I look forward to hearing from you.
Sincerely,
Songying Ouyang, Ph.D.
Professor
Key Laboratory of OptoElectronic Science and Technology for Medicine of the Ministry of Education,
Biomedical Research Center of South China,
College of Life Science,
Fujian Normal University,
Fuzhou 350117, Fujian, China
Tel: 86-0591-22868199
E-mail: ouyangsy@fjnu.edu.cn
Responses to the comments of Reviewer #1
- It is not clear what was the aim of the study, as Authors did not formulate it. Based on the title, it may be supposed that Authors intended to present novel knowledge which we have in 2023 while compared with the previous studies/ previous reports. However, there is a very serious problem, as Authors do not present what is new in our knowledge, but reproduce well known information (which were already presented in a number of papers), but novel knowledge is not even mentioned.
Response:
We are very grateful to the reviewers for their valuable comments. The purpose of the study has been revised in the manuscript (lines 57-71).. In terms of presenting new knowledge, our aim was to provide a comprehensive summary of the latest drug developments and advancements, focusing on mainstream targets. It is possible that some basic concepts related to the targets, such as distribution and mechanism, may have been duplicated in the published manuscript. However, in order to ensure the inclusion of the most up-to-date information, we have made efforts to review and incorporate the most recent studies in the new manuscript (lines 163-170, lines 229-234, lines 362-364, lines 472-477, lines 546-552), thereby addressing any shortcomings in the original manuscript. As for the newly found potential targets in the past two years, we selected some of them for review, TSC22D4 and NLK have been proposed in the original manuscript, and we added the content about GSN, ChREBP, and IsletMI Cs in the new manuscript (lines 769-804). In the added miRNAs, diet and exercise we also try to summarize the latest content (lines 758-767, lines 832-841). The details are as follows:
(1) The aim of the study(lines 57-71)
However, to date, no cure for diabetes is available. Treatments such as medications are available to retain the blood glucose level as close to normal as possible, thus delaying or preventing diabetes-related health problems. Further, due to the existence of drug resistance, long-term treatment limited to a particular drug against a specific target may gradually become ineffective. Moreover, different individuals may have different underlying diseases, and treatment against a certain target may not be applicable to everyone due to their variability. In addition, multi-target drugs have generally been shown to be more effective than single-target drugs. Therefore, there is a need to continuously identify new targets and develop efficient and safe therapeutic drugs. Meanwhile, due to the cumbersome process of long-term drug administration, emerging technologies such as stem cell therapy and CRISPR therapy can be tried to solve such difficulties. Based on this, this article reviews the mechanism of action and the development of the latest therapeutic agents for the popular targets of T2DM in recent years, as well as the emerging target-based therapies and the new potential therapeutic targets that have emerged in the last 3 years, to provide a theoretical basis for further advancing the targeted therapy of T2DM.
(2) Novel knowledge
① Lines 163-170:
The FDA has warned about the increased risk of heart failure events with DPP-4 inhibitors, but an updated meta-analysis showed no significant risk of heart failure events [25]. When it was used in conjunction with metformin, the incidence of cardio-vascular events was not significantly different from that of SGLT-2 inhibitors with cardiovascular benefits [26]. The incidence of major adverse cardiovascular events is also lower when compared to that of sulfonylureas with similar efficacy [27].
② Lines 229-234:
Li et al. [41] continuously optimized the benzene ring structure of HBK001 and synthesized the hydrochloride form of HBK001 (HBK001 hydrochloride) with an oxadiazole side chain structure, which improved the DPP-4 inhibitory effect and moderate agonism of GPR119, showing improved bioavailability and hypoglycemic effects in vivo. However, it showed moderate inhibition of hERG channels, probably due to its high lipophilicity.
③ Lines 362-364:
Liu et al. [66] demonstrated the antidiabetic activity of cyclic enol ether terpenoids isolated from Patrinia scabiosaefolia, which was achieved by upregulating the phosphorylation level of AKT.
④ Lines 472-477:
Queen [83] et al. attempted to utilize gene therapy to treat diabetes, minus the cumbersome need for regular administration of FGF 21 analogs. They gave insulin-resistant mice a single low dose of an adenoviral vector encoding FGF 21 and showed that the method consistently counteracted insulin resistance without adverse effects, as well as reduced body weight and inflammation.
⑤ Lines 546-552:
Casertano [95] et al. demonstrated that Avarone, a sesquiterpene quinone extracted from marine sponge Dysidea avara, can improve both insulin sensitivity and mitochondrial activity and is also a tight binding inhibitor of aldose reductase, which prevents diabetic retinal complications. Ali [96] et al. demonstrated that ursonic acid extracted from Artemisia montana also inhibits the expression of PTP1B, which activates the GLUT4 in the PI3K/AKT signaling pathway and increases peripheral glucose up-take.
⑥ Lines 758-767:
Therefore, miRNAs can be used as biomarkers for diabetes prediction [152]. A me-ta-analysis showed that miR-29a-3p, miR-221-3p, miR-126-3p,miR-26a-5p, miR-503-5p, miR-100-5p, miR-101-3p, mIR-103a-3p, miR-122-5p, miR-199a-3p, miR-30b-5p, miR-130a-3p, miR-143-3p, miR-145-5p, miR-19a-3p, and miR-311-3p (in order of importance) fulfill the criteria for biomarker selection [153]. However, there is heterogeneity in the expression of miRNAs, and some studies have found sex variability in their use as markers, which may be due to the following mechanisms: estrogen regulates the transcription and processing of miRNAs, incomplete X-chromosome inactivation leading to biallelic expression of miRNAs, and miRNA expression is regulated by epigenetics [154].
⑦ Lines 769-804:
6.5. Glucose-sensitive neurons (GSN)
The hypothalamus is a central part of the central system involved in the regulation of energy metabolism, and its dysfunction can lead to systemic metabolic disorders such as obesity and diabetes. GSNs are present in the middle and base layers of the hypothalamus, which are characterized by glucose sensing, but their responses to changes in glucose levels are not identical. Glucose-excited neurons (GE) show in-creased activity at high glucose levels and decreased activity at low glucose levels, whereas glucose-inhibited neurons (GI) show the opposite characteristics [155]. A re-cent study has shown that GSNs hold promise as new targets for antidiabetic therapy. In this study, FGF4 was administered to T2DM mice through the lateral ventricle, and a single administration produced durable glucose-controlling effects for up to 7 weeks or more. Exploration of the mechanism revealed that GSNs and their highly expressed FGFR1 are the key target cells and preferred receptor subtypes that mediate the regulation of persistent glucose homeostasis by FGF4. Additionally, further knockdown experiments showed that GI plays a key role [156].
6.6. Carbohydrate response element binding protein(ChREBP)
ChREBP is a glucose-responsive transcription factor with two isoforms, ChREBPα and ChREBPβ. Deletion of ChREBP has been shown to prevent glucotoxicity and glucose-mediated β-cell death. However, recent studies have found that ChREBP is required for glucose-stimulated β-cell proliferation. Whereas overexpression of ChREBPβ leads to glucose toxicity and its subsequent death in β-cells, overexpression of ChREBPα enhances glucose-stimulated β-cell proliferation as it stimulates the Nrf2 antioxidant pathway, thereby preventing oxidative damage [157]. Therefore, the development of ChREBPα selective activators could be attempted for the treatment of diabetes.
6.7. Islet microexons (IsletMICs)
Microexons are DNA sequences that encode proteins, approximately 3 to 27 nucleotides long, that can be selectively spliced in neurons, microglia, embryonic stem cells, and cancer cells to produce cell type-specific protein isoforms [158]. A recent study found that IsletMICs can be spliced into pancreatic islet cell mRNAs and play an important role in islet function and glycemic control. IsletMICs are regulated by the RNA-binding protein SRRM3, and both SRRM3 and IsletMICs are induced by elevated glucose levels. If depletion of human, rat β-cell lines, and mouse pancreatic islets of SRRM3, or the use of antisense oligonucleotides to inhibit specific IsletMICs, can lead to inappropriate insulin secretion, it is the IsletMICs that are at low levels in patients with T2DM, so attempts can be made to upregulate the levels of IsletMICs to obtain therapy [159].
⑧ Lines 832-841:
Whereas ω-3 appears to have different efficacy, a previous large-scale prospective study showed that ω-3 reduces the risk of T2DM [169], but the recent mendelian randomized study showed that ω-3 can act through two different clusters of genetic variants, with one set of variants ameliorating insulin resistance while the other contributing to β-cell dysfunction and increasing T2DM risk [170].
- From 146 references only 1 (one!) was published in 2023, while 76 of them was published until 2020. Taking into account how quick the knowledge is broadened, the referred studies are old (if Authors intended to present ‘advances’ in the studied area).
Response:
We wish to express our sincere gratitude for the invaluable suggestions provided by the reviewer, alongside our profound remorse for the error committed on our behalf. Our failure to consider the rapid expansion of knowledge is a clear manifestation of negligence, leading to an excessive inclusion of pre-2020 literature. In the revised manuscript, we have rectified this issue by updating the literature. Out of the initial 76 pre-2020 sources, only 3 have been retained, while the remaining have been replaced with literature from the 2020-2023 period, with 18 sources specifically from 2023.
- The serious flaw of the presented manuscript is associated with the fact, that it presents a highly subjective review, not a systematic review. While the systematic review has a key role for broadening knowledge, the other reviews don’t have such role.
Response:
Please accept our deepest thanks for the valuable suggestions offered by the reviewer, as well as our deepest apologies for the error we made. We very much agree with the reviewer that systematic analysis can bring readers more comprehensive and objective knowledge. This study aims to tackle a particular issue through an exhaustive compilation of pertinent studies, employing standardized methodologies to critically assess and analyze each one individually, ultimately leading to an impartial and comprehensive conclusion. Nevertheless, in contrast to alternative forms of review writing, this approach necessitates a more extensive and rigorous literature search and data analysis expertise, as well as a greater time investment. Since the paper had been revised in advance, finishing this task within the confined timeframe of over 10 days proved challenging.
- This problem is noted while we compare the ‘advances’ described by Authors with really novel drug targets for type 2 diabetes, which are presented by other Authors, e.g. In the study like https://www.ncbi.nlm.nih.gov/pmc/articles/PMC8869656/ - even if published in 2020, Authors of this study present a number of novel therapeutic options – emerging targets, which Authors of the presented study do not even mention.
Response:
We express our sincere gratitude to the reviewers for their invaluable feedback. The literature provided by the reviewers has been thoroughly examined. Although sulphonylureas, which are associated with insulin secretion, were mentioned as common targets in the article, we have chosen not to include them in our review due to their limited research in the past five years. Conversely, we have already reviewed the remaining targets, namely GLP-1, DPP-4, and SGLT-2. Furthermore, we have also previously reviewed the emerging targets PTP1B and PPAR. Furthermore, our review has discussed the AKT signaling pathway, wherein FoxO1 serves as a crucial molecular node, and its mechanism of action has been succinctly elucidated within the paper. Our literature search has indicated a scarcity of studies pertaining to FFA2 and FFA3 in comparison to FFA1 and FFA4, thus limiting our review to solely encompass FFA1 and FFA4. The remaining targets either did not meet the criteria of our screening process or were predominantly focused on the treatment of diabetic complications, thereby precluding their inclusion in our analysis.
- Taking into account, that the Materials and methods section is not presented (it should be added), without any specific information, it is hard to understand which areas were included, and consequently which studies were included into review and why. Authors did not present any key words, which were used during literature search, inclusion and exclusion criteria of references, information about the procedure of literature search conducted by them, number of chosen references, as well as information if some of them were excluded from the review and on the basis of which criteria. As a number of recent publications that are related to the issue were not included, it is a serious problem.
Response:
Thanks to the reviewers for their valuable comments. We agree with the reviewers that the criteria used in the literature search should be included in the article to give the reader a more complete picture of the screen conditions under which our review was written. We have added this section in the new manuscript (lines 72-77), as described below:
- Method
In this paper, PubMed, Scopus, Web of Science, and Google Scholar databases were used as sources, and the terms "target, drug, mechanism, diabetes mellitus, and insulin resistance" were used as the main keywords for searching to collect relevant literature from 2018–2023 for analysis, and then, based on the specifics of the reviewed targets as the keywords, the relevant literature from the past 3 years was collected for discussion.
- Authors do not present the current and comprehensive knowledge associated with the issue. It is associated with the fact that they did not include some important issues, while other were included even if they are not so crucial (or not novel).
Response:
We would like to extend our heartfelt appreciation for the invaluable suggestion put forth by the reviewer. We have now added new elements to the new manuscript (lines 747-864) to make the article more comprehensive. The details are as follows:
6.4. microRNA(miRNA)
miRNAs are a class of non-coding single-stranded RNA molecules encoded by endogenous genes with a length of approximately 22 nucleotides, which play a regulatory role on target mRNAs by destabilizing and inhibiting the translation of target mRNAs. A single miRNA can regulate the expression of multiple target mRNAs, and each mRNA can be regulated by multiple miRNAs. Cells can also release miRNAs in free form or in complex with extracellular vesicles, which can be taken up by other types of cells, thus mediating cell-to-cell actions [148]. miRNAs can influence insulin signaling by affecting the expression of INSR and IRS-1, the translocation of GLUT4, and the activity of PI3K, and they can also influence insulin secretion by regulating β cell metabolic stress and proliferation, survival, as well as regulating GSIS and improving insulin sensitivity [148-151]. Therefore, miRNAs can be used as biomarkers for diabetes prediction [152]. A meta-analysis showed that miR-29a-3p, miR-221-3p, miR-126-3p,miR-26a-5p, miR-503-5p, miR-100-5p, miR-101-3p, mIR-103a-3p, miR-122-5p, miR-199a-3p, miR-30b-5p, miR-130a-3p, miR-143-3p, miR-145-5p, miR-19a-3p, and miR-311-3p (in order of importance) fulfill the criteria for biomarker selection [153]. However, there is heterogeneity in the expression of miRNAs, and some studies have found sex variability in their use as markers, which may be due to the following mechanisms: estrogen regulates the transcription and processing of miRNAs, incomplete X-chromosome inactivation leading to biallelic expression of miRNAs, and miRNA expression is regulated by epigenetics [154]. To date, no miRNA-based anti-diabetic therapies have been approved by the FDA [148].
6.5. Glucose-sensitive neurons (GSN)
The hypothalamus is a central part of the central system involved in the regulation of energy metabolism, and its dysfunction can lead to systemic metabolic disorders such as obesity and diabetes. GSNs are present in the middle and base layers of the hypothalamus, which are characterized by glucose sensing, but their responses to changes in glucose levels are not identical. Glucose-excited neurons (GE) show in-creased activity at high glucose levels and decreased activity at low glucose levels, whereas glucose-inhibited neurons (GI) show the opposite characteristics [155]. A re-cent study has shown that GSNs hold promise as new targets for antidiabetic therapy. In this study, FGF4 was administered to T2DM mice through the lateral ventricle, and a single administration produced durable glucose-controlling effects for up to 7 weeks or more. Exploration of the mechanism revealed that GSNs and their highly expressed FGFR1 are the key target cells and preferred receptor subtypes that mediate the regulation of persistent glucose homeostasis by FGF4. Additionally, further knockdown experiments showed that GI plays a key role [156].
6.6. Carbohydrate response element binding protein(ChREBP)
ChREBP is a glucose-responsive transcription factor with two isoforms, ChREBPα and ChREBPβ. Deletion of ChREBP has been shown to prevent glucotoxicity and glucose-mediated β-cell death. However, recent studies have found that ChREBP is required for glucose-stimulated β-cell proliferation. Whereas overexpression of ChREBPβ leads to glucose toxicity and its subsequent death in β-cells, overexpression of ChREBPα enhances glucose-stimulated β-cell proliferation as it stimulates the Nrf2 antioxidant pathway, thereby preventing oxidative damage [157]. Therefore, the development of ChREBPα selective activators could be attempted for the treatment of diabetes.
6.7. Islet microexons (IsletMICs)
Microexons are DNA sequences that encode proteins, approximately 3 to 27 nucleotides long, that can be selectively spliced in neurons, microglia, embryonic stem cells, and cancer cells to produce cell type-specific protein isoforms [158]. A recent study found that IsletMICs can be spliced into pancreatic islet cell mRNAs and play an important role in islet function and glycemic control. IsletMICs are regulated by the RNA-binding protein SRRM3, and both SRRM3 and IsletMICs are induced by elevated glucose levels. If depletion of human, rat β-cell lines, and mouse pancreatic islets of SRRM3, or the use of antisense oligonucleotides to inhibit specific IsletMICs, can lead to inappropriate insulin secretion, it is the IsletMICs that are at low levels in patients with T2DM, so attempts can be made to upregulate the levels of IsletMICs to obtain therapy [159].
- Diet and exercise based therapy
Although there is a genetic predisposition to T2DM, the onset and progression of the disease are more influenced by the environment, with unhealthy diet and lack of exercise being some of the factors. Despite continuous attempts to develop new drugs, side effects are common, and even the first-line drug, metformin, can cause nausea and gastric distension. There is also compliance with medication, and symptoms may not improve once the medication is stopped. Therefore, if we can intervene through good lifestyle habits such as proper diet and exercise, and medication is used as adjunctive therapy, it will be a lower cost and healthier treatment.
Diet is recognized as a cornerstone in the prevention and treatment of abnormal glucose metabolism and insulin resistance [160], and the ADA released an updated consensus report on diabetes nutrition therapy in 2019 [161]. The report emphasized that there should not be a "one-size-fits-all" dietary pattern for people with diabetes; however, individualized choices should be made because no dietary pattern is completely healthy and can be applied to everyone, but regardless of the pattern, it is generally based on three basic principles: eat more non-starchy vegetables (e.g., broccoli, kale, mushrooms, etc.); eat less sugar and refined grains; choose natural foods over highly processed foods. The ketogenic diet is one of the more researched dietary pat-terns today, which is a low-calorie, low-carbohydrate, high-fat, protein-appropriate diet that uses ketone bodies as a source of energy and mimics the fasting state without causing the negative consequences of starvation [162]. A meta-analysis showed that the ketogenic diet was effective in improving glycemic parameters, body weight, and lipid profile of patients [163]. Randomized controlled trials have also shown that a ketogenic diet reduces HbA1c and triglyceride levels in patients [164]. However, it should be noted that ketogenic diets may lead to metabolic ketoacidosis [165]. Besides this, the mediterranean diet, low-sugar diet, and medium-carbohydrate diet have shown better control of HbA1c and fasting blood glucose levels in patients [166]. Some micronutrients such as ω-3, polyphenols, and vitamin D have also shown a correlation with T2DM. Vitamin D reduces the risk of the disease in prediabetic patients [167], while polyphenols are anti-inflammatory and antioxidant, reduce β-cell apoptosis as well as inhibit α-glucosidase, modulate the intestinal microbiota, and improve adipose tissue metabolism [168]. Whereas ω-3 appears to have different efficacy, a previous large-scale prospective study showed that ω-3 reduces the risk of T2DM [169], but the recent mendelian randomized study showed that ω-3 can act through two different clusters of genetic variants, with one set of variants ameliorating insulin resistance while the other contributing to β-cell dysfunction and increasing T2DM risk [170]. It is widely believed that potentially harmful diets should be completely eliminated and beneficial diets should be completely followed, but diets actually have complexities. Excessive fiber intake can lead to belly bloating and diarrhea; abstainers have a higher prevalence of the disease than moderate drinkers; indiscriminate combinations and large stacks of macronutrients are counterproductive [161].
Exercise is also a cornerstone for maintaining glucose homeostasis in patients with T2DM, as it increases overall energy consumption and specifically promotes the uptake of peripheral glucose by skeletal muscle in an insulin-independent manner [171], which attenuates endoplasmic reticulum stress caused by overproduction of insulin by β-cells due to overnutrition. Moreover, reducing glucose entry into β-cells for aerobic metabolism directly reduces ROS production and attenuates oxidative stress [172]. A recent large cohort study showed that appropriate exercise increases cardiorespiratory fitness, and higher cardiorespiratory fitness reduces T2DM risk [173]. The American College of Sports Medicine updated the Consensus Statement on exercise in patients with T2DM in 2022 [174]. It is mentioned here that the improvement in systemic insulin sensitivity in patients can last from 2 to 72 hours after any exercise and that the reduction in blood glucose is closely related to the duration and intensity of exercise. Regular aerobic exercise improves glycemic management, reduces time to daily hyperglycemia, and lowers HbA1c levels in adult patients with T2DM, as demonstrated by a meta-analysis on the Baduanjin [175]. For elderly patients, resistance training and flexibility training are more suitable for their physical condition. Breaking up sedentary behavior by doing "little and often" daily physical activity can moderately lower postprandial blood glucose, especially in patients with insulin resistance and high BMI. Exercise safety should also be emphasized. Exercise should be prohibited in the case of blood glu-cose >250 mg/dL, and patients with insulin or insulin prosecretors should carry fast carbohydrates such as candy to prevent hypoglycemia [174].
In conclusion, a healthy diet and regular exercise can reduce the risk of developing T2DM and benefit patients with T1DM [176,177].
- Last but not least, for the review articles proper referring the source of information is crucial, while Authors do not do it properly, suggesting that their review just reproduced observations by other authors of review articles. It may be noted from the first sentence, as Authors wrote that “In 2018 the American Diabetes Association classified diabetes mellitus into four major categories […] [1].” – as they mentioned American Diabetes Association, it may have been supposed that they referred American Diabetes Association. But they did not. They referred the review article “Type 2 Diabetes Mellitus: A Review of Multi-Target Drugs” (https://www.mdpi.com/1420-3049/25/8/1987) which presented state of knowledge for 2020. In this study the position by American Diabetes Association from 2018 was referred, but since this moment the novel position of American Diabetes Association was published in 2022 (https://pubmed.ncbi.nlm.nih.gov/34964875/). Authors not only refer improper reference, but also present false information about 2018.
Response:
We express our gratitude for your valuable comment and extend our sincere apologies for any confusion that may have arisen. We have made corrections in the new manuscript (lines 21-26, lines 916-917) as follows:
- Lines 29-34
In 2022, the American Diabetes Association (ADA) categorized diabetes into four main types based on different pathogenesis: type 1 diabetes, type 2 diabetes, specific types of diabetes caused by other reasons (e.g., monogenic diabetes syndromes, diseases of the exocrine pancreas, drug- or chemically-induced diabetes), and gestational diabetes, with type 2 diabetes accounting for 90–95% of the cases [1].
- Lines 914-915
- American Diabetes Association Professional Practice Committee 6. Glycemic Targets: Standards of Medical Care in Diabetes—2022. Diabetes Care. 2022, 45, S83–S96. DOI:10.2337/dc22-S006.

Reviewer 2 Report
The revision work is interesting, however, it lacks some points of fundamental importance:
- Type II diabetes, although it may have a genetic component, is established due to incorrect nutrition and lack of physical activity; this must be underlined in a significant way, as one cannot think of treating type II diabetes only with pharmacological therapy; drugs should be seen as support, not the solution.
- The part of the microbiota should be avoided as it is even more in the field of hypotheses
- A section dealing with miRNAs could be considered (e.g. 10.3390/ph14121257 or also the role of vitamin A derivatives 10.2174/1389557515666150709112822
- A paragraph should be added, not just 2 lines, to underline the need for adequate physical activity (which is beneficial even in the case of type I diabetes, see 10.14814/phy2.15740) and an equally careful diet.
- Finally, some micronutrients such as omega3 and polyphenols could also be mentioned
It needs revision
Author Response
August 11, 2023
Prof. Dr. Maurizio Battino
Editor-in-Chief
Mr. Bryan Li,
Assistant Editor
International Journal of Molecular Sciences
Dear Editor,
I wish to re-submit the manuscript titled “Advances in Research on Type 2 Diabetes Mellitus Targets and Therapeutic Agents.” The manuscript ID is ijms-2532957.
Please accept our sincere appreciation for the valuable suggestions and insightful perspectives provided by you and the reviewers. These astute recommendations have greatly enhanced the manuscript.
Attached is the revised version of our manuscript. In the following pages are our point-by-point responses to each of the comments of the reviewers. Revisions in the text are highlighted by the utilization of the color red. We hope that the revisions in the manuscript and our accompanying responses would be sufficient to make our manuscript suitable for publication in International Journal of Molecular Sciences.
Thank you for your consideration. I look forward to hearing from you.
Sincerely,
Songying Ouyang, Ph.D.
Professor
Key Laboratory of OptoElectronic Science and Technology for Medicine of the Ministry of Education,
Biomedical Research Center of South China,
College of Life Science,
Fujian Normal University,
Fuzhou 350117, Fujian, China
Tel: 86-0591-22868199
E-mail: ouyangsy@fjnu.edu.cn
Responses to the comments of Reviewer #2
- Type II diabetes, although it may have a genetic component, is established due to incorrect nutrition and lack of physical activity; this must be underlined in a significant way, as one cannot think of treating type II diabetes only with pharmacological therapy; drugs should be seen as support, not the solution.
Response:
We would like to extend our heartfelt appreciation for the invaluable suggestion provided by the reviewer. We have emphasized the importance of nutrition and exercise and the fact that medications can only assist in treatment in the new manuscript (lines 803-810), as described below:
Although there is a genetic predisposition to T2DM, the onset and progression of the disease are more influenced by the environment, with unhealthy diet and lack of exercise being some of the factors. Despite continuous attempts to develop new drugs, side effects are common, and even the first-line drug, metformin, can cause nausea and gastric distension. There is also compliance with medication, and symptoms may not improve once the medication is stopped. Therefore, if we can intervene through good lifestyle habits such as proper diet and exercise, and medication is used as adjunctive therapy, it will be a lower cost and healthier treatment.
- The part of the microbiota should be avoided as it is even more in the field of hypotheses
Response:
We express our gratitude to the reviewers for their invaluable feedback. The length of this section has been reduced by providing a concise summary (lines 672-746). Given the increasing papers of gut microbiota research in recent years, a portion of it has been retained in our study.
- A section dealing with miRNAs could be considered (e.g. 10.3390/ph14121257 or also the role of vitamin A derivatives 10.2174/1389557515666150709112822)
Response:
We express our gratitude to the reviewers for their invaluable feedback. It is evident that in recent times, Omics analysis has gained considerable popularity, with a particular focus on the investigation of non-coding RNAs, specifically miRNAs. These molecules have been extensively examined within the realm of diabetes mellitus research and have demonstrated notable research implications. Hence, we have opted to present a comprehensive overview of miRNA content (lines 747-767). The specifics are outlined as follows:
6.4. microRNA(miRNA)
miRNAs are a class of non-coding single-stranded RNA molecules encoded by endogenous genes with a length of approximately 22 nucleotides, which play a regulatory role on target mRNAs by destabilizing and inhibiting the translation of target mRNAs. A single miRNA can regulate the expression of multiple target mRNAs, and each mRNA can be regulated by multiple miRNAs. Cells can also release miRNAs in free form or in complex with extracellular vesicles, which can be taken up by other types of cells, thus mediating cell-to-cell actions [148]. miRNAs can influence insulin signaling by affecting the expression of INSR and IRS-1, the translocation of GLUT4, and the activity of PI3K, and they can also influence insulin secretion by regulating β cell metabolic stress and proliferation, survival, as well as regulating GSIS and improving insulin sensitivity [148-151]. Therefore, miRNAs can be used as biomarkers for diabetes prediction [152]. A meta-analysis showed that miR-29a-3p, miR-221-3p, miR-126-3p,miR-26a-5p, miR-503-5p, miR-100-5p, miR-101-3p, mIR-103a-3p, miR-122-5p, miR-199a-3p, miR-30b-5p, miR-130a-3p, miR-143-3p, miR-145-5p, miR-19a-3p, and miR-311-3p (in order of importance) fulfill the criteria for biomarker selection [153]. However, there is heterogeneity in the expression of miRNAs, and some studies have found sex variability in their use as markers, which may be due to the following mechanisms: estrogen regulates the transcription and processing of miRNAs, incomplete X-chromosome inactivation leading to biallelic expression of miRNAs, and miRNA expression is regulated by epigenetics [154]. To date, no miRNA-based anti-diabetic therapies have been approved by the FDA [148].
- A paragraph should be added, not just 2 lines, to underline the need for adequate physical activity (which is beneficial even in the case of type I diabetes, see 10.14814/phy2.15740) and an equally careful diet.
Response:
We express our gratitude to the reviewers for their invaluable feedback. We acknowledge the significance of incorporating diet and exercise as cost-effective measures for the prevention and management of diabetes in everyday life. Consequently, we have included a dedicated section addressing this topic in the revised manuscript (lines 811-864). The specific details are outlined below:
Diet is recognized as a cornerstone in the prevention and treatment of abnormal glucose metabolism and insulin resistance [160], and the ADA released an updated consensus report on diabetes nutrition therapy in 2019 [161]. The report emphasized that there should not be a "one-size-fits-all" dietary pattern for people with diabetes; however, individualized choices should be made because no dietary pattern is completely healthy and can be applied to everyone, but regardless of the pattern, it is generally based on three basic principles: eat more non-starchy vegetables (e.g., broccoli, kale, mushrooms, etc.); eat less sugar and refined grains; choose natural foods over highly processed foods. The ketogenic diet is one of the more researched dietary pat-terns today, which is a low-calorie, low-carbohydrate, high-fat, protein-appropriate diet that uses ketone bodies as a source of energy and mimics the fasting state without causing the negative consequences of starvation [162]. A meta-analysis showed that the ketogenic diet was effective in improving glycemic parameters, body weight, and lipid profile of patients [163]. Randomized controlled trials have also shown that a ketogenic diet reduces HbA1c and triglyceride levels in patients [164]. However, it should be noted that ketogenic diets may lead to metabolic ketoacidosis [165]. Besides this, the mediterranean diet, low-sugar diet, and medium-carbohydrate diet have shown better control of HbA1c and fasting blood glucose levels in patients [166]. Some micronutrients such as ω-3, polyphenols, and vitamin D have also shown a correlation with T2DM. Vitamin D reduces the risk of the disease in prediabetic patients [167], while polyphenols are anti-inflammatory and antioxidant, reduce β-cell apoptosis as well as inhibit α-glucosidase, modulate the intestinal microbiota, and improve adipose tissue metabolism [168]. Whereas ω-3 appears to have different efficacy, a previous large-scale prospective study showed that ω-3 reduces the risk of T2DM [169], but the recent mendelian randomized study showed that ω-3 can act through two different clusters of genetic variants, with one set of variants ameliorating insulin resistance while the other contributing to β-cell dysfunction and increasing T2DM risk [170]. It is widely believed that potentially harmful diets should be completely eliminated and beneficial diets should be completely followed, but diets actually have complexities. Excessive fiber intake can lead to belly bloating and diarrhea; abstainers have a higher prevalence of the disease than moderate drinkers; indiscriminate combinations and large stacks of macronutrients are counterproductive [161].
Exercise is also a cornerstone for maintaining glucose homeostasis in patients with T2DM, as it increases overall energy consumption and specifically promotes the uptake of peripheral glucose by skeletal muscle in an insulin-independent manner [171], which attenuates endoplasmic reticulum stress caused by overproduction of insulin by β-cells due to overnutrition. Moreover, reducing glucose entry into β-cells for aerobic metabolism directly reduces ROS production and attenuates oxidative stress [172]. A recent large cohort study showed that appropriate exercise increases cardiorespiratory fitness, and higher cardiorespiratory fitness reduces T2DM risk [173]. The American College of Sports Medicine updated the Consensus Statement on exercise in patients with T2DM in 2022 [174]. It is mentioned here that the improvement in systemic insulin sensitivity in patients can last from 2 to 72 hours after any exercise and that the reduction in blood glucose is closely related to the duration and intensity of exercise. Regular aerobic exercise improves glycemic management, reduces time to daily hyperglycemia, and lowers HbA1c levels in adult patients with T2DM, as demonstrated by a meta-analysis on the Baduanjin [175]. For elderly patients, resistance training and flexibility training are more suitable for their physical condition. Breaking up sedentary behavior by doing "little and often" daily physical activity can moderately lower postprandial blood glucose, especially in patients with insulin resistance and high BMI. Exercise safety should also be emphasized. Exercise should be prohibited in the case of blood glucose >250 mg/dL, and patients with insulin or insulin prosecretors should carry fast carbohydrates such as candy to prevent hypoglycemia [174].
In conclusion, a healthy diet and regular exercise can reduce the risk of developing T2DM and benefit patients with T1DM [176,177].
- Finally, some micronutrients such as omega3 and polyphenols could also be mentioned
Response:
We express our gratitude to the reviewers for their invaluable feedback. Despite constituting a minor proportion of the body's nutrient requirements, micronutrients play a pivotal role in orchestrating various physiological processes. Consequently, we have incorporated this particular section into the revised manuscript (lines 828-836). The details are as follows:
Some micronutrients such as ω-3, polyphenols, and vitamin D have also shown a cor-relation with T2DM. Vitamin D reduces the risk of the disease in prediabetic patients [167], while polyphenols are anti-inflammatory and antioxidant, reduce β-cell apoptosis as well as inhibit α-glucosidase, modulate the intestinal microbiota, and improve adipose tissue metabolism [168]. Whereas ω-3 appears to have different efficacy, a previous large-scale prospective study showed that ω-3 reduces the risk of T2DM [169], but the recent mendelian randomized study showed that ω-3 can act through two different clusters of genetic variants, with one set of variants ameliorating insulin resistance while the other contributing to β-cell dysfunction and increasing T2DM risk [170].

Round 2
Reviewer 2 Report
The authors improved the manuscript following my suggestion.
Just a second English review is needed.
- Line 826 keto acidosis could occur in type I diabetes, but not in type II if the diet is well designed. The part on KD could be a bit expanded, highlighting the anti-inflammatory and antioxidant action of this kind of diet (10.3390/antiox8080269 and 10.1016/j.eplepsyres.2020.106454 for example)
It needs some revision.
Author Response
August 17, 2023
Prof. Dr. Maurizio Battino
Editor-in-Chief
Mr. Bryan Li,
Assistant Editor
International Journal of Molecular Sciences
Dear Editor,
I wish to re-submit the manuscript titled “Advances in Research on Type 2 Diabetes Mellitus Targets and Therapeutic Agents.” The manuscript ID is ijms-2532957.
We would like to express our profound gratitude for the invaluable suggestions and enlightening perspectives contributed by both yourself and the reviewers. These discerning recommendations have significantly augmented the quality of the manuscript.
Attached is the revised version of our manuscript. In the following pages are our point-by-point responses to each of the comments of the reviewers. Revisions in the text are highlighted by the utilization of the color red. We hope that the revisions in the manuscript and our accompanying responses would be sufficient to make our manuscript suitable for publication in International Journal of Molecular Sciences.
Thank you for your consideration. I look forward to hearing from you.
Sincerely,
Songying Ouyang, Ph.D.
Professor
Key Laboratory of OptoElectronic Science and Technology for Medicine of the Ministry of Education,
Biomedical Research Center of South China,
College of Life Science,
Fujian Normal University,
Fuzhou 350117, Fujian, China
Tel: 86-0591-22868199
E-mail: ouyangsy@fjnu.edu.cn
Responses to the comments of Reviewer #2
- The authors improved the manuscript following my suggestion.
Respond:
We express our sincere gratitude to the reviewer for the invaluable suggestions.
- Just a second English review is needed.
Response:
We would like to express our profound appreciation to the reviewer for their invaluable suggestions. The manuscript underwent editing by Editage, a professional language editing company. The specific modifications are as follows:
(1)Lines 14-16
Diabetes mellitus is a chronic multifaceted disease with multiple potential complications, the treatment of which can only delay and prolong the terminal stage of the disease, i.e., type 2 diabetes mellitus (T2DM).
(2)Lines 18-20
single-target drugs are gradually failing to meet the therapeutic requirements owing to the individual variability, diversity of pathogenesis, and organismal resistance.
(3)Lines 22-26
This article presents an overview of the mechanisms of action and the development of the latest therapeutic agents targeting T2DM in recent years. It also discusses emerging target-based therapies and new potential therapeutic targets that have emerged within the last 3 years. The aim is to pro-vide a theoretical basis for further advancement in targeted therapies for T2DM.
(4)Lines 38-42
Newly confirmed models of pathogenesis suggest that nutritional loading causes a chronic increase in insulin secretion, leading to hyperinsulinemia, which in turn triggers insulin resistance until β-cell failure, ultimately challenging the conventional belief that insulin resistance precedes the onset of hyperinsulinemia in the development of overt T2DM.
(5)Lines 43-45
537 million people (more than 75% of whom are from low- and middle-income countries) were diagnosed with diabetes in 2021
(6)Lines 49-52
such as Alzheimer's disease, becoming the leading causes of death for people with diabetes in some regions; further, patients with diabetes are at a high risk of infections, primarily including foot, respiratory, urinary tract, and postoperative infections
(7)Lines 56-58
furthermore, there is evidence of a mutual exacerbation between these conditions and the underlying disease [3]. Therefore, the effective treatment of patients with diabetes remains warranted.
(8)Lines 59-65
However, to date, cure for diabetes is lacking, while treatments such as medications are available to maintain the blood glucose level as close to normal levels as possible, thus delaying or preventing diabetes-related health problems. Further, owing to drug resistance, long-term treatment limited to a particular drug against a specific target may gradually become ineffective. Different individuals may have different underlying diseases, and treatment against a certain target may not be applicable to everyone because of the variability. In addition, multi-target drugs are more effective than single-target drugs.
(9)Lines 67-73
Meanwhile, owing to the cumbersome process of long-term drug administration, emerging technologies such as stem cell therapy and CRISPR therapy can be attempted to resolve such difficulties. Accordingly, this review presents an overview of the mechanisms of action and the development of the latest therapeutic agents targeting T2DM in recent years. It also discusses emerging target-based therapies and new potential therapeutic targets that have emerged within the last 3 years. The aim is to provide a theoretical basis for further advancement in targeted therapies for T2DM.
(10)Lines 74-79
- Methods
PubMed, Scopus, Web of Science, and Google Scholar databases were used as sources, and the terms "target, drug, mechanism, diabetes mellitus, and insulin resistance" were used as the main keywords to collect relevant literature from 2018–2023 for analysis, and, based on the specifics of the reviewed targets as the keywords, the relevant literature from the past 3 years was collected for discussion.
- Lines 88-91
GLP-1 inhibits glucagon secretion along with stimulating insulin secretion during hyperglycemia [8], reducing appetite, delaying gastric emptying by stimulating vagal afferents, and activating hindbrain PKA, MAPK and inhibiting hindbrain 5'-AMPK phosphorylation
- Lines 99-100
Drugs that target GLP-1 are mainly its receptor agonists; these drugs have replaced insulin as the main line of treatment for diabetes.
- Lines 109-113
GIP receptor agonists are not used as single-target drugs to treat patients with diabetes, because after GIPR is activated, in addition to the lipogenic effect of insulin, GIP can recruit chylomicrons, enhance the activity of lipoprotein lipase, promote the hydrolysis of dietary triglycerides and the release of free fatty acids, and promote lipid storage, thus leading to obesity
- Lines 123-124
Tirzepatide was the first US Food and Drug Administration (FDA)-approved GLP-1 and GIP receptor dual agonist drug.
- Lines 128-129
Furthermore, Tirzepatide may prevent cardiovascular disease and atherosclerosis, among others
(16)Lines 131-133
Instead, the incidence of gastrointestinal events increase following treatment with Tirzepatide, possibly owing to the suboptimal dose in clinical trials
(17)Lines 150-151
while inhibiting glucagon secretion from α cells and preventing hypoglycemic risk due to the negative feedback regulation of glucagon by GIP in hypoglycemia.
- Lines 154-157
The current gliptin series of drugs for DPP-4 targets are small molecules, which are cheap to manufacture, easily absorbed, highly stable, have low mutagenicity, and do not require special storage conditions.
- Lines 169-170
The incidence of major adverse cardiovascular events is also lower when compared with that of sulfonylureas with similar efficacy
- Lines 174-176
GPCRs are the largest family of membrane proteins in the human genome and the most common targets for FDA-approved drugs owing to their high drug-containing properties and the multiple physiological pathways mediated in the human body
(21)Lines 197-198
however, neither are available for clinical use because of poor efficacy or severe side effects
(22)Lines 200-202
However, TAK-875 was terminated at this stage owing to hepatotoxicity issues, prob-ably due to its own metabolites, which affect bile acid and bilirubin homeostasis, leading to hyperbilirubinemia and cholestatic hepatotoxicity
- Lines 207-212
SCO-267 was the first full agonist to be revealed in a clinical trial profile and can be administered orally once daily. In Phase I clinical trials, SCO-267 was safe and well tolerated after single and multiple doses in both healthy adults and patients with diabetes, was effective in improving glucose tolerance in patients with diabetes, and did not cause hypoglycemia; it is currently being developed for Phase II trials
- Lines 218-219
GPR119, which is a class A GPCR [32], is highly expressed in pancreatic β-cells and intestinal endocrine K and L cells activated by endogenous cannabinoid-like compounds
- Lines 222-224
At the same time, GPR119 activation also increased glucagon secretion, which may reduce the risk of medically induced hypoglycemia in patients with diabetes under-going intensive insulin therapy
- Lines 231-236
Li et al. [41] have previously optimized the benzene ring structure of HBK001 and synthesized the hydrochloride form of HBK001 (HBK001 hydrochloride) with an oxadiazole side chain structure, which improved the DPP-4 inhibitory effect and moderate agonism of GPR119, showing improved bioavailability and hypoglycemic effects in vivo, but moderate inhibition of hERG channels, probably due to its high lipophilicity.
- Lines 237-239
GPR120, another class A GPCR [32], is activated by long-chain fatty acids and is widely expressed in intestinal endocrine L, K, and I cells, pancreatic β-, δ-, and α-cells, adipocytes, and macrophages
- Lines 249-252
However, in general, due to the lipophilic nature of the ligands of GPCRs and the biased nature of signal transduction, as well as the unresolved crystal structures of GPR120 and GPR119, the development of single-targeted drugs with high efficacy and safety requires much research.
- Lines 258-260
GK has a higher Km value compared with the other three subtypes of the hexokinase family (I–III) and its activity is not inhibited by its product glucose-6-phosphate
- Line 280
VGCC, voltage-gated calcium channel
- Lines 301-304
Dyslipidemia and fatty liver production may be due to the overactivation of GKs in hepatocytes by GKAs, leading to excessive accumulation of G-6-P and elevated levels of acetyl coenzyme A, ultimately increasing the influx of fatty acids and triglycerides, as well as hepatic de novo lipogenesis
- Lines 317-320
AKT, also known as PKB, is a Ser/Thr kinase that contains three structural do-mains, as follows: a central kinase structural domain with specificity for substrate protein Ser/Thr residues, a Pleckstrin homology structural domain (PH) that mediates membrane recruitment, and a carboxyterminal regulatory structural domain
- Lines 326-330
Various growth factors, cytokines, and hormones first activate the receptor tyrosine kinase, while PI3K, the upstream molecule of AKTs, receives the signal that catalyzes the conversion of phosphatidylinositol 4,5-bisphosphate (PIP2) to phosphatidylinositol 3,4,5-trisphosphate (PIP3), which binds to and activate PDK1, thereby translocating it to the cell membrane.
- Lines 359-360
The use of herbal medicines to activate AKT and thus achieve hypoglycemia is a hot research spot.
- Lines 368-369
TSC22D4 is a member of the TSC22 protein family that comprises the subtypes TSC22D1, TSC22D2, and TSC22D3
- Lines 371-375
The N-terminal of TSC22D4 contains two domains R1 and R2, and the R2 domain and TSC boxes are connected by a highly conserved disordered region (Figure 6), which contains different post-translational modification sites, mediating the interaction of TSC22D4 with different proteins
- Lines 378-388
The expression of LCN13 was significantly up-regulated by knocking down TSC22D4, and the phosphorylation level of AKT, a key protein of the insulin pathway, was also up-regulated, while the expression of gluconeogenesis-related genes was down-regulated. Furthermore, knockdown of LCN13+TSC22D4 in db/db mice up-regulated fasting glucose levels and down-regulated glucose tolerance compared with TSC22D4 knockdown mice. This suggests that LCN13 is at least partially involved in the improvement of blood glucose levels brought about by the TSC22D4 knockdown [68]. Recent studies have further elucidated the glucose regulation mechanism of TSC22D4. TSC22D4 interacts with AKT1, which affects the phosphorylation levels of AKT and its downstream proteins while influencing the expression of genes related to the glucose metabolic pathway.
- Lines 433-436
The fibroblast growth factor (FGF) superfamily includes 22 family members, which are divided into seven subfamilies, most of which are present in the extracellular matrix and are involved in the regulation of physiological processes by paracrine or autocrine means using acetyl heparin sulfate as a cofactor
- Lines 449-460
FGF21 administration promoted weight loss and improved insulin sensitivity in multiple preclinical models [79]. However, human-natural FGF21 (hFGF21) is not suitable for clinical use owing to its poor pharmacokinetic properties; its short half-life of 0.5–2 h may be partly due to its small molecular weight (19.5 kDa), which increases its filtration rate by the glomerulus, and partly due to its susceptibility to cleavage by proteases in plasma or its inactivation by aggregation into insoluble proteins under physiological conditions [75,80]. hFGF21 is subject to DPP-4 and fibroblast-activating protein (FAP) cleavage, with DPP-4 responsible for cleavage at the Pro-2 and Pro-4 sites and FAP responsible for cleavage at the Pro-171 site, which is followed by the C-terminus, a KLB binding site; thus, cleavage of Pro-171 inhibits its binding to KLB, which results in loss of activity [75,77]. Therefore, attempts have been made to develop hFGF21 analogs or use gene therapy and stem cell therapy to promote clinical treatment of this target.
- Lines 466-474
Meanwhile, in diabetic mice, FGF21SS had higher biological activity; in inflammatory adipocytes, it inhibited the nuclear factor-κB (NF-κB) signaling pathway, reversed insulin resistance induced by inflammatory factors, and had good hypoglycemic, weight loss, and serum insulin-lowering abilities [82]. Queen et al. [83] attempted to utilize gene therapy to treat diabetes, minus the cumbersome need for regular administration of FGF 21 analogs. The authors treated insulin-resistant mice with a single low dose of an adenoviral vector encoding FGF 21 and showed that the method consistently counteracted insulin resistance without adverse effects, as well as reduced body weight and inflammation.
- Line 481
Stem cell therapy has been applied to the treatment of T2DM.
- Lines 483-485
The authors showed that this method significantly reduced blood glucose and body weight, increased insulin sensitivity, and improved the lipid profile.
- Lines 487-492
In addition, the hypoglycemic function of many analogs is inactivated in human treatment, probably due to the low abundance of brown adipose tissue in humans compared to that in rodents, which is an important target organ for FGF21. Further-more, the duration of existing clinical trials of FGF21-based therapies is short, whereas T2DM is a chronic metabolic disease; therefore, longer-term treatment is needed to assess the associated efficacy and safety issues
- Lines 497-504
PTP1B belongs to the intracellular PTPs expressed in various cells and tissues, and its encoding gene, PTPN1, is located in a region associated with insulin resistance and obesity; hence, it is involved in the regulation of insulin metabolic pathways [86]. PTP1B consists of three structural domains, as follows: the N-terminal catalytic domain, the regulatory structural domain, and the C-terminal domain that targets the endoplasmic reticulum membrane [87]. The N-terminal catalytic domain active site contains Arg that can generate a positive charge, and Cys, which has nucleophilic activity, both of which are essential for the catalytic action of PTP1B.
- Lines 506-509
Further, phosphorylation of Tyr residues on the catalytic structural domain of PTP1B is one of the mechanisms regulating its activity, which is active when its Tyr 66 residue is phosphorylated, whereas it is inactivated when its Ser 50 residue is phosphorylated
- Lines 517-519
The active site of PTP1B is highly positively charged; hence, an effective inhibitor should be anionically charged or strongly polarized, whereas these properties of the inhibitor weaken its membrane permeability and limit oral bioavailability
- Lines 526-528
their druggability is assessed to be much higher than that when targeting the active sites; however, better inhibition and selectivity are observed if both sites are targeted simultaneously
- Lines 537-538
however, all the trials were eventually discontinued because of adverse side effects and low specificity
- Lines 539-541
Currently, researchers are paying close attention to natural products. Natural products isolated from natural plants have good biocompatibility, low side effects, synergistic effects on a wide range of diseases, and are of low cost with great medicinal value
- Lines 560-562
SGLT-2 is in an overexpressed state in patients with T2DM, which enhances glucose uptake by the kidneys and increases blood glucose levels.
- Lines 564-566
Moreover, the regulatory mechanism of SGLT-2 inhibitors is insulin-independent; hence, there is generally no risk of hypoglycemia, and it protects β-cells from glucose toxicity
- Lines 569-572
SGLT-2 inhibitors have more effects that can also prevent diabetic complications. SGLT-2 inhibitors have a lower all-cause mortality rate compared with DPP-4 inhibitors and GLP-1R agonists, especially in terms of cardiovascular mortality and hospitalization for heart failure
- Lines 587-591
For example, Empagliflozin has the most additional benefits among them besides the hypoglycemic effect and is the first hypoglycemic drug in the world to reduce the risk of cardiovascular death in patients with T2DM. It is now included in the World Health Organization Essential Medicines List, increasing the accessibility and affordability of this drug in the future
- Lines 593-603
Henagliflozin was the first original SGLT-2 inhibitor in China, which has under-gone a 10-year long research run and was approved for marketing at the end of 2021. Henagliflozin is structurally optimized, resulting in better chemical stability, drug solubility, and high receptor selectivity, with the SGLT-2/SGLT-1 receptor selectivity ratio up to 1,823.53 [115,116]. The phase III clinical study showed that Henagliflozin has a good hypoglycemic effect on both fasting and postprandial glucose, and when combined with metformin, it provides long-term efficacy (the effect can continue for 28 weeks after stopping treatment) [116,117]. In addition, Henagliflozin provided multiple benefits such as weight loss and blood pressure reduction and has a good safety and tolerability profile, with an incidence of adverse events comparable to that of the placebo with no additional risk of hypoglycemia and urinary tract infection [116].
- Lines 610-612
PPARβ/δ is expressed in all organs but highly expressed in skeletal muscle, adipose tissue, and the heart and is mainly involved in the regulation of fatty acid oxidation and blood glucose levels
- Lines 616-626
PPARγ mainly regulates adipocyte differentiation [118], lipid storage, and insulin sensitivity [118,121,122]. PPARs have four major functional domains, as follows: the N-terminal non-ligand-dependent transcriptional activation domain (A/B structural domain), where phosphorylation of this region leads to inhibition of the transcriptional activation function of PPARs; the DNA-binding structural domain, also known as the C structural domain, responsible for binding to the peroxisome proliferator response element (PPRE) on the promoter of PPAR target genes; the D structural domain, which is the docking site for various cofactors; and the large Y-shaped hydrophobic binding pocket formed at the C-terminus, which is the ligand-binding structural domain (E/F structural domain) that confers the ability to bind endogenous or exogenous lipophilic ligands to PPARs
- Lines 652-657
Chiglitazar Sodium is the first PPAR full agonist independently developed in China and is approved for the treatment of T2DM. It was included in the national health insurance program in January 2023. Owing to the simultaneous moderate activation of three functionally different but overlapping PPAR subtypes, Chiglitazar Sodium attains the ability to not only selectively alter the expression of a series of genes related to insulin sensitivity
- Lines 660-664
The results of phase III clinical trials of Chiglitazar Sodium showed that it significantly reduced HbA1c levels by 1.52%, and its hypoglycemic efficacy was stable for up to 52 weeks, with improvements in fasting and 2-h postprandial glucose compared with that with sitagliptin. It also significantly improved insulin sensitivity, protected islet secretion, and reduced triglyceride levels.
- Lines 674-680
T2DM are mainly conditionally pathogenic, such as Bacteroides caccae, Escherichia coli, and Eggerthella lenta, and they reduce abundance of butyrate-producing bacteria, including Eubacterium rectale, Faecalibacterium prausnitzii, and Roseburia intestinalis [131]. Although the specific mechanisms by which gut microbiota affect the development of T2DM are complex, some of the clearer mechanisms involve the short-chain fatty acid (SCFA), bile acid (BA), branched-chain amino acid (BCAA), endotoxin-intestinal barrier, imidazole propionate (ImP), and aryl hydrocarbon receptor (AhR) theories
- Lines 690-693
Butyrate can bind to GPR41 and GPR43 to protect β-cells from oxidative stress, inhibit deacetylation of histones, down-regulate the expression of genes related to gluconeogenesis in the liver, and bind to GLP-1R to promote insulin secretion
- Lines 695-702
The main gut microbiota are involved in the synthesis, modification, and signal transduction of BAs. These microbiota have the ability to produce bile salt hydrolases, enzymes that convert primary BAs into secondary BAs. This process is crucial for the biological functions of BAs [139]. Primary BAs can bind to and activate the farnesoid X receptor (FXR), which improves glucose tolerance and increases insulin sensitivity [134,138]. Secondary BAs bind to Takeda-G-protein receptor-5 (TGR-5) and induce GLP-1 secretion, thereby promoting insulin secretion [129].
- Lines 712-715
LPS is recognized by innate toll-like receptors, which later activate the nuclear factor kappa-B (NF-κB) signaling pathway, promoting the release of pro-inflammatory factors, thereby leading to inflammation. This in turn triggers β-cell destruction, insulin resistance, and increased intestinal wall permeability [130,134].
- Lines 720-723
ImP is a histidine-derived metabolite elevated in prediabetic and T2DM patients [132], and has been shown to impair insulin signaling by activating mTORC1 and in-creasing Ser phosphorylation of IRS-1 [142]. Furthermore, it impairs the effects of metformin by inhibiting AMPK activity [143].
- Lines 733-735
Mixtures of probiotics and prebiotics or combinations of both with antidiabetic drugs have a superior efficacy than probiotics or prebiotics alone [144,145].
- Lines 738-739
However, there are safety concerns; one case of severe bacteremia and one death have been reported.
- Lines 749-751
Cells can release miRNAs in free form or in complex with extracellular vesicles, which can be taken up by other types of cells, thus mediating cell-to-cell actions [148].
- Lines 752-755
furthermore, they influence insulin secretion by regulating β cell metabolic stress and proliferation, survival, as well as regulating GSIS and improving insulin sensitivity [148-151]. Therefore, miRNAs are potential biomarkers for diabetes prediction [152].
- Lines 759-763
However, there is heterogeneity in the expression of miRNAs, and some studies have found sex variability in their use as markers, which may be owing to the following mechanisms: estrogen regulates the transcription and processing of miRNAs, incomplete X-chromosome inactivation leading to biallelic expression of miRNAs, and regulation of miRNA expression by epigenetics [154].
- Lines 777-778
Additionally, further knockdown experiments confirmed that GI plays a key role [156].
- Lines 782-787
However, recent findings have suggested that ChREBP is required for glucose-stimulated β-cell proliferation. Overexpression of ChREBPβ leads to glucose toxicity and its subsequent death of β-cells, while overexpression of ChREBPα enhances glucose-stimulated β-cell proliferation as it stimulates the Nrf2 antioxidant pathway, thereby preventing oxidative damage [157]. Therefore, the development of ChREBPα selective activators are potential treatment against diabetes.
- Lines 789-791
Microexons are DNA sequences that encode proteins, approximately 3 to 27 nucleotides long, which can be selectively spliced in neurons, microglia, embryonic stem cells, and cancer cells to produce cell type-specific protein isoforms [158].
- Lines 794-798
If depletion of SRRM3 in human and rat β-cell lines, as well as mouse pancreatic islets, or the use of antisense oligonucleotides to inhibit specific IsletMICs, can result in in-appropriate insulin secretion, it suggests that IsletMICs are present at low levels in patients with T2DM; hence, up-regulation of IsletMICs levels is a potential therapeutic strategy [159].
- Lines 800-807
Although there is a genetic predisposition to T2DM, the onset and progression of the disease are influenced more by the environment, with unhealthy diet and lack of exercise being major factors. Despite continuous attempts to develop new drugs, side effects remain challenging, and even the first-line drug, metformin, can cause nausea and gastric distension. There is also the issue of compliance with medication, and symptoms may not improve once the medication is stopped. Therefore, intervention through good lifestyle habits such as proper diet and exercise, with medication as adjunctive therapy, will be a lower-cost and healthier treatment option.
- Lines 808-810
Diet is a cornerstone in the prevention and treatment of abnormal glucose metabolism and insulin resistance [160]. The ADA released an updated consensus report on diabetes nutrition therapy in 2019 [161].
- Lines 813-815
but regardless of the pattern, it is generally based on three basic principles, as follows: consuming more non-starchy vegetables (such as broccoli, kale and mushrooms), less sugar and refined grains, and choosing natural foods over highly processed foods.
- Lines 831-833
Besides, the Mediterranean diet, low-sugar diet, and medium-carbohydrate diet have shown better control of HbA1c and fasting blood glucose levels in patients [170].
- Lines 835-846
while polyphenols have anti-inflammatory and antioxidant properties, reduce β-cell apoptosis, inhibit α-glucosidase, modulate the intestinal microbiota, and improve adipose tissue metabolism [172]. While ω-3 appears to have different efficacy, a previous large-scale prospective study showed that ω-3 reduces the risk of T2DM [173]; however, a recent mendelian randomized study showed that ω-3 can act through two different clusters of genetic variants, with one set of variants ameliorating insulin resistance while the other contributing to β-cell dysfunction and increasing T2DM risk [174]. It is believed that potentially harmful diets should be completely eliminated and beneficial diets should be completely followed, but diets have complexities. Excessive fiber intake can lead to belly bloating and diarrhea; abstainers have a higher prevalence of the disease than moderate drinkers; and indiscriminate combinations and large stacks of macronutrients are counterproductive [161].
- Lines 856-858
Improvements in systemic insulin sensitivity in patients can last from 2 to 72 h after exercise and the reduction in blood glucose is closely related to the duration and in-tensity of exercise.
- Lines 861-867
Breaking up sedentary behavior by doing "little and often" daily physical activity can moderately lower postprandial blood glucose levels, especially in patients with insulin resistance and high BMI. Exercise safety should also be emphasized. In cases where blood glucose levels exceed 250 mg/dL, exercise should be prohibited. Additionally, patients who are on insulin or insulin secretagogues should have fast-acting carbohydrates, such as candy, readily available to prevent hypoglycemia [178].
- Lines 871-873
Diabetes is a public health issue of global concern as a chronic multifaceted disease that remains incurable and is accompanied by many complications such as cardiovascular disease and emotional-cognitive dysfunction.
- Lines 874-876
Concurrently, owing to the large population base, the prevalence of diabetes in China will only increase as the population aging process continues to accelerate.
- Lines 880-882
such as Semaglutide, which has weight loss, hypoglycemic, and cardiovascular protective effects, and as a long-acting drug, it can be injected once a week, eliminating the tediousness of daily dosing.
- Lines 883-884
Stem cell therapy is also making its debut in the field of diabetes.
- Lines 888-893
Despite the continuously proposed treatment strategies, China still has the highest number of patients with diabetes in the world. Ultimately, individual differences, diversity of pathogenesis, and drug resistance have a significant influence. Different patients may have different underlying diseases, and drugs targeting a certain target may not be suitable for everyone. For instance, metformin is not suitable for patients with heart failure or renal dysfunction.
- Lines 899-900
Therefore, future studies must investigate the changes in the omics of lesioned cells and the specific metabolic pathways they affect.
- Lines 902-904
In conclusion, the treatment and prevention of diabetes requires a concerted effort by all parties involved and not just scientist/physicians.
- - Line 826 keto acidosis could occur in type I diabetes, but not in type II if the diet is well designed. The part on KD could be a bit expanded, highlighting the anti-inflammatory and antioxidant action of this kind of diet (10.3390/antiox8080269 and 10.1016/j.eplepsyres.2020.106454 for example)
Response:
We are very grateful to the reviewers for their valuable comments. We have added content on the anti-inflammatory and antioxidant effects of KD in the new manuscript (lines 822-831). The details are as follows:
Furthermore, it has been observed that diabetes leads to dysregulation in the anti-inflammatory and oxidative stress pathways. However, the implementation of a KD has shown potential in mitigating these detrimental effects. The KD exerts its anti-inflammatory effects through various mechanisms such as adenosine, ketone bodies, mTOR pathways, PPARγ, NLRP3 inflammasome, and gut microbiota. Additionally, the KD exhibits antioxidant properties by improving mitochondrial dynamics, regulating miRNA expression associated with oxidative stress, and enhancing the pentose phosphate pathway and related antioxidant defense systems [165-168]. However, it should be noted that a specific ketogenic diet plan may lead to metabolic ketoacidosis if it is not well designed [169].
Reference:
- Koh, S.; Dupuis, N.; Auvin, S. Ketogenic diet and Neuroinflammation. Epilepsy Res. 2020, 167, 106454. DOI:10.1016/j.eplepsyres.2020.106454.
- Paoli, A.; Cerullo, G. Investigating the Link between Ketogenic Diet, NAFLD, Mitochondria, and Oxidative Stress: A Narrative Review. Antioxidants (Basel). 2023, 12, 1065. DOI:10.3390/antiox12051065.
- Cannataro, R.; Caroleo, M.C.; Fazio, A.; La Torre, C.; Plastina, P.; Gallelli, L.; Lauria, G.; Cione, E. Ketogenic Diet and microRNAs Linked to Antioxidant Biochemical Homeostasis. Antioxidants(Basel). 2019, 8, 269. DOI:10.3390/antiox8080269.
- Peng, F.; Zhang, Y.-H.; Zhang, L.; Yang, M.; Chen, C.; Yu, H.; Li, T. Ketogenic diet attenuates post-cardiac arrest brain injury by upregulation of pentose phosphate pathway-mediated antioxidant defense in a mouse model of cardiac arrest. Nutrition. 2022, 103–104, 111814. DOI:10.1016/j.nut.2022.111814.
- Churuangsuk, C.; Hall, J.; Reynolds, A.; Griffin, S.J.; Combet, E.; Lean, M.E.J. Diets for weight management in adults with type 2 diabetes: an umbrella review of published meta-analyses and systematic review of trials of diets for diabetes remission. Diabetologia. 2022, 65, 14–36. DOI:10.1007/s00125-021-05577-2.
August 17, 2023
Prof. Dr. Maurizio Battino
Editor-in-Chief
Mr. Bryan Li,
Assistant Editor
International Journal of Molecular Sciences
Dear Editor,
I wish to re-submit the manuscript titled “Advances in Research on Type 2 Diabetes Mellitus Targets and Therapeutic Agents.” The manuscript ID is ijms-2532957.
We would like to express our profound gratitude for the invaluable suggestions and enlightening perspectives contributed by both yourself and the reviewers. These discerning recommendations have significantly augmented the quality of the manuscript.
Attached is the revised version of our manuscript. In the following pages are our point-by-point responses to each of the comments of the reviewers. Revisions in the text are highlighted by the utilization of the color red. We hope that the revisions in the manuscript and our accompanying responses would be sufficient to make our manuscript suitable for publication in International Journal of Molecular Sciences.
Thank you for your consideration. I look forward to hearing from you.
Sincerely,
Songying Ouyang, Ph.D.
Professor
Key Laboratory of OptoElectronic Science and Technology for Medicine of the Ministry of Education,
Biomedical Research Center of South China,
College of Life Science,
Fujian Normal University,
Fuzhou 350117, Fujian, China
Tel: 86-0591-22868199
E-mail: ouyangsy@fjnu.edu.cn
Responses to the comments of Reviewer #2
- The authors improved the manuscript following my suggestion.
Respond:
We express our sincere gratitude to the reviewer for the invaluable suggestions.
- Just a second English review is needed.
Response:
We would like to express our profound appreciation to the reviewer for their invaluable suggestions. The manuscript underwent editing by Editage, a professional language editing company. The specific modifications are as follows:
(1)Lines 14-16
Diabetes mellitus is a chronic multifaceted disease with multiple potential complications, the treatment of which can only delay and prolong the terminal stage of the disease, i.e., type 2 diabetes mellitus (T2DM).
(2)Lines 18-20
single-target drugs are gradually failing to meet the therapeutic requirements owing to the individual variability, diversity of pathogenesis, and organismal resistance.
(3)Lines 22-26
This article presents an overview of the mechanisms of action and the development of the latest therapeutic agents targeting T2DM in recent years. It also discusses emerging target-based therapies and new potential therapeutic targets that have emerged within the last 3 years. The aim is to pro-vide a theoretical basis for further advancement in targeted therapies for T2DM.
(4)Lines 38-42
Newly confirmed models of pathogenesis suggest that nutritional loading causes a chronic increase in insulin secretion, leading to hyperinsulinemia, which in turn triggers insulin resistance until β-cell failure, ultimately challenging the conventional belief that insulin resistance precedes the onset of hyperinsulinemia in the development of overt T2DM.
(5)Lines 43-45
537 million people (more than 75% of whom are from low- and middle-income countries) were diagnosed with diabetes in 2021
(6)Lines 49-52
such as Alzheimer's disease, becoming the leading causes of death for people with diabetes in some regions; further, patients with diabetes are at a high risk of infections, primarily including foot, respiratory, urinary tract, and postoperative infections
(7)Lines 56-58
furthermore, there is evidence of a mutual exacerbation between these conditions and the underlying disease [3]. Therefore, the effective treatment of patients with diabetes remains warranted.
(8)Lines 59-65
However, to date, cure for diabetes is lacking, while treatments such as medications are available to maintain the blood glucose level as close to normal levels as possible, thus delaying or preventing diabetes-related health problems. Further, owing to drug resistance, long-term treatment limited to a particular drug against a specific target may gradually become ineffective. Different individuals may have different underlying diseases, and treatment against a certain target may not be applicable to everyone because of the variability. In addition, multi-target drugs are more effective than single-target drugs.
(9)Lines 67-73
Meanwhile, owing to the cumbersome process of long-term drug administration, emerging technologies such as stem cell therapy and CRISPR therapy can be attempted to resolve such difficulties. Accordingly, this review presents an overview of the mechanisms of action and the development of the latest therapeutic agents targeting T2DM in recent years. It also discusses emerging target-based therapies and new potential therapeutic targets that have emerged within the last 3 years. The aim is to provide a theoretical basis for further advancement in targeted therapies for T2DM.
(10)Lines 74-79
- Methods
PubMed, Scopus, Web of Science, and Google Scholar databases were used as sources, and the terms "target, drug, mechanism, diabetes mellitus, and insulin resistance" were used as the main keywords to collect relevant literature from 2018–2023 for analysis, and, based on the specifics of the reviewed targets as the keywords, the relevant literature from the past 3 years was collected for discussion.
- Lines 88-91
GLP-1 inhibits glucagon secretion along with stimulating insulin secretion during hyperglycemia [8], reducing appetite, delaying gastric emptying by stimulating vagal afferents, and activating hindbrain PKA, MAPK and inhibiting hindbrain 5'-AMPK phosphorylation
- Lines 99-100
Drugs that target GLP-1 are mainly its receptor agonists; these drugs have replaced insulin as the main line of treatment for diabetes.
- Lines 109-113
GIP receptor agonists are not used as single-target drugs to treat patients with diabetes, because after GIPR is activated, in addition to the lipogenic effect of insulin, GIP can recruit chylomicrons, enhance the activity of lipoprotein lipase, promote the hydrolysis of dietary triglycerides and the release of free fatty acids, and promote lipid storage, thus leading to obesity
- Lines 123-124
Tirzepatide was the first US Food and Drug Administration (FDA)-approved GLP-1 and GIP receptor dual agonist drug.
- Lines 128-129
Furthermore, Tirzepatide may prevent cardiovascular disease and atherosclerosis, among others
(16)Lines 131-133
Instead, the incidence of gastrointestinal events increase following treatment with Tirzepatide, possibly owing to the suboptimal dose in clinical trials
(17)Lines 150-151
while inhibiting glucagon secretion from α cells and preventing hypoglycemic risk due to the negative feedback regulation of glucagon by GIP in hypoglycemia.
- Lines 154-157
The current gliptin series of drugs for DPP-4 targets are small molecules, which are cheap to manufacture, easily absorbed, highly stable, have low mutagenicity, and do not require special storage conditions.
- Lines 169-170
The incidence of major adverse cardiovascular events is also lower when compared with that of sulfonylureas with similar efficacy
- Lines 174-176
GPCRs are the largest family of membrane proteins in the human genome and the most common targets for FDA-approved drugs owing to their high drug-containing properties and the multiple physiological pathways mediated in the human body
(21)Lines 197-198
however, neither are available for clinical use because of poor efficacy or severe side effects
(22)Lines 200-202
However, TAK-875 was terminated at this stage owing to hepatotoxicity issues, prob-ably due to its own metabolites, which affect bile acid and bilirubin homeostasis, leading to hyperbilirubinemia and cholestatic hepatotoxicity
- Lines 207-212
SCO-267 was the first full agonist to be revealed in a clinical trial profile and can be administered orally once daily. In Phase I clinical trials, SCO-267 was safe and well tolerated after single and multiple doses in both healthy adults and patients with diabetes, was effective in improving glucose tolerance in patients with diabetes, and did not cause hypoglycemia; it is currently being developed for Phase II trials
- Lines 218-219
GPR119, which is a class A GPCR [32], is highly expressed in pancreatic β-cells and intestinal endocrine K and L cells activated by endogenous cannabinoid-like compounds
- Lines 222-224
At the same time, GPR119 activation also increased glucagon secretion, which may reduce the risk of medically induced hypoglycemia in patients with diabetes under-going intensive insulin therapy
- Lines 231-236
Li et al. [41] have previously optimized the benzene ring structure of HBK001 and synthesized the hydrochloride form of HBK001 (HBK001 hydrochloride) with an oxadiazole side chain structure, which improved the DPP-4 inhibitory effect and moderate agonism of GPR119, showing improved bioavailability and hypoglycemic effects in vivo, but moderate inhibition of hERG channels, probably due to its high lipophilicity.
- Lines 237-239
GPR120, another class A GPCR [32], is activated by long-chain fatty acids and is widely expressed in intestinal endocrine L, K, and I cells, pancreatic β-, δ-, and α-cells, adipocytes, and macrophages
- Lines 249-252
However, in general, due to the lipophilic nature of the ligands of GPCRs and the biased nature of signal transduction, as well as the unresolved crystal structures of GPR120 and GPR119, the development of single-targeted drugs with high efficacy and safety requires much research.
- Lines 258-260
GK has a higher Km value compared with the other three subtypes of the hexokinase family (I–III) and its activity is not inhibited by its product glucose-6-phosphate
- Line 280
VGCC, voltage-gated calcium channel
- Lines 301-304
Dyslipidemia and fatty liver production may be due to the overactivation of GKs in hepatocytes by GKAs, leading to excessive accumulation of G-6-P and elevated levels of acetyl coenzyme A, ultimately increasing the influx of fatty acids and triglycerides, as well as hepatic de novo lipogenesis
- Lines 317-320
AKT, also known as PKB, is a Ser/Thr kinase that contains three structural do-mains, as follows: a central kinase structural domain with specificity for substrate protein Ser/Thr residues, a Pleckstrin homology structural domain (PH) that mediates membrane recruitment, and a carboxyterminal regulatory structural domain
- Lines 326-330
Various growth factors, cytokines, and hormones first activate the receptor tyrosine kinase, while PI3K, the upstream molecule of AKTs, receives the signal that catalyzes the conversion of phosphatidylinositol 4,5-bisphosphate (PIP2) to phosphatidylinositol 3,4,5-trisphosphate (PIP3), which binds to and activate PDK1, thereby translocating it to the cell membrane.
- Lines 359-360
The use of herbal medicines to activate AKT and thus achieve hypoglycemia is a hot research spot.
- Lines 368-369
TSC22D4 is a member of the TSC22 protein family that comprises the subtypes TSC22D1, TSC22D2, and TSC22D3
- Lines 371-375
The N-terminal of TSC22D4 contains two domains R1 and R2, and the R2 domain and TSC boxes are connected by a highly conserved disordered region (Figure 6), which contains different post-translational modification sites, mediating the interaction of TSC22D4 with different proteins
- Lines 378-388
The expression of LCN13 was significantly up-regulated by knocking down TSC22D4, and the phosphorylation level of AKT, a key protein of the insulin pathway, was also up-regulated, while the expression of gluconeogenesis-related genes was down-regulated. Furthermore, knockdown of LCN13+TSC22D4 in db/db mice up-regulated fasting glucose levels and down-regulated glucose tolerance compared with TSC22D4 knockdown mice. This suggests that LCN13 is at least partially involved in the improvement of blood glucose levels brought about by the TSC22D4 knockdown [68]. Recent studies have further elucidated the glucose regulation mechanism of TSC22D4. TSC22D4 interacts with AKT1, which affects the phosphorylation levels of AKT and its downstream proteins while influencing the expression of genes related to the glucose metabolic pathway.
- Lines 433-436
The fibroblast growth factor (FGF) superfamily includes 22 family members, which are divided into seven subfamilies, most of which are present in the extracellular matrix and are involved in the regulation of physiological processes by paracrine or autocrine means using acetyl heparin sulfate as a cofactor
- Lines 449-460
FGF21 administration promoted weight loss and improved insulin sensitivity in multiple preclinical models [79]. However, human-natural FGF21 (hFGF21) is not suitable for clinical use owing to its poor pharmacokinetic properties; its short half-life of 0.5–2 h may be partly due to its small molecular weight (19.5 kDa), which increases its filtration rate by the glomerulus, and partly due to its susceptibility to cleavage by proteases in plasma or its inactivation by aggregation into insoluble proteins under physiological conditions [75,80]. hFGF21 is subject to DPP-4 and fibroblast-activating protein (FAP) cleavage, with DPP-4 responsible for cleavage at the Pro-2 and Pro-4 sites and FAP responsible for cleavage at the Pro-171 site, which is followed by the C-terminus, a KLB binding site; thus, cleavage of Pro-171 inhibits its binding to KLB, which results in loss of activity [75,77]. Therefore, attempts have been made to develop hFGF21 analogs or use gene therapy and stem cell therapy to promote clinical treatment of this target.
- Lines 466-474
Meanwhile, in diabetic mice, FGF21SS had higher biological activity; in inflammatory adipocytes, it inhibited the nuclear factor-κB (NF-κB) signaling pathway, reversed insulin resistance induced by inflammatory factors, and had good hypoglycemic, weight loss, and serum insulin-lowering abilities [82]. Queen et al. [83] attempted to utilize gene therapy to treat diabetes, minus the cumbersome need for regular administration of FGF 21 analogs. The authors treated insulin-resistant mice with a single low dose of an adenoviral vector encoding FGF 21 and showed that the method consistently counteracted insulin resistance without adverse effects, as well as reduced body weight and inflammation.
- Line 481
Stem cell therapy has been applied to the treatment of T2DM.
- Lines 483-485
The authors showed that this method significantly reduced blood glucose and body weight, increased insulin sensitivity, and improved the lipid profile.
- Lines 487-492
In addition, the hypoglycemic function of many analogs is inactivated in human treatment, probably due to the low abundance of brown adipose tissue in humans compared to that in rodents, which is an important target organ for FGF21. Further-more, the duration of existing clinical trials of FGF21-based therapies is short, whereas T2DM is a chronic metabolic disease; therefore, longer-term treatment is needed to assess the associated efficacy and safety issues
- Lines 497-504
PTP1B belongs to the intracellular PTPs expressed in various cells and tissues, and its encoding gene, PTPN1, is located in a region associated with insulin resistance and obesity; hence, it is involved in the regulation of insulin metabolic pathways [86]. PTP1B consists of three structural domains, as follows: the N-terminal catalytic domain, the regulatory structural domain, and the C-terminal domain that targets the endoplasmic reticulum membrane [87]. The N-terminal catalytic domain active site contains Arg that can generate a positive charge, and Cys, which has nucleophilic activity, both of which are essential for the catalytic action of PTP1B.
- Lines 506-509
Further, phosphorylation of Tyr residues on the catalytic structural domain of PTP1B is one of the mechanisms regulating its activity, which is active when its Tyr 66 residue is phosphorylated, whereas it is inactivated when its Ser 50 residue is phosphorylated
- Lines 517-519
The active site of PTP1B is highly positively charged; hence, an effective inhibitor should be anionically charged or strongly polarized, whereas these properties of the inhibitor weaken its membrane permeability and limit oral bioavailability
- Lines 526-528
their druggability is assessed to be much higher than that when targeting the active sites; however, better inhibition and selectivity are observed if both sites are targeted simultaneously
- Lines 537-538
however, all the trials were eventually discontinued because of adverse side effects and low specificity
- Lines 539-541
Currently, researchers are paying close attention to natural products. Natural products isolated from natural plants have good biocompatibility, low side effects, synergistic effects on a wide range of diseases, and are of low cost with great medicinal value
- Lines 560-562
SGLT-2 is in an overexpressed state in patients with T2DM, which enhances glucose uptake by the kidneys and increases blood glucose levels.
- Lines 564-566
Moreover, the regulatory mechanism of SGLT-2 inhibitors is insulin-independent; hence, there is generally no risk of hypoglycemia, and it protects β-cells from glucose toxicity
- Lines 569-572
SGLT-2 inhibitors have more effects that can also prevent diabetic complications. SGLT-2 inhibitors have a lower all-cause mortality rate compared with DPP-4 inhibitors and GLP-1R agonists, especially in terms of cardiovascular mortality and hospitalization for heart failure
- Lines 587-591
For example, Empagliflozin has the most additional benefits among them besides the hypoglycemic effect and is the first hypoglycemic drug in the world to reduce the risk of cardiovascular death in patients with T2DM. It is now included in the World Health Organization Essential Medicines List, increasing the accessibility and affordability of this drug in the future
- Lines 593-603
Henagliflozin was the first original SGLT-2 inhibitor in China, which has under-gone a 10-year long research run and was approved for marketing at the end of 2021. Henagliflozin is structurally optimized, resulting in better chemical stability, drug solubility, and high receptor selectivity, with the SGLT-2/SGLT-1 receptor selectivity ratio up to 1,823.53 [115,116]. The phase III clinical study showed that Henagliflozin has a good hypoglycemic effect on both fasting and postprandial glucose, and when combined with metformin, it provides long-term efficacy (the effect can continue for 28 weeks after stopping treatment) [116,117]. In addition, Henagliflozin provided multiple benefits such as weight loss and blood pressure reduction and has a good safety and tolerability profile, with an incidence of adverse events comparable to that of the placebo with no additional risk of hypoglycemia and urinary tract infection [116].
- Lines 610-612
PPARβ/δ is expressed in all organs but highly expressed in skeletal muscle, adipose tissue, and the heart and is mainly involved in the regulation of fatty acid oxidation and blood glucose levels
- Lines 616-626
PPARγ mainly regulates adipocyte differentiation [118], lipid storage, and insulin sensitivity [118,121,122]. PPARs have four major functional domains, as follows: the N-terminal non-ligand-dependent transcriptional activation domain (A/B structural domain), where phosphorylation of this region leads to inhibition of the transcriptional activation function of PPARs; the DNA-binding structural domain, also known as the C structural domain, responsible for binding to the peroxisome proliferator response element (PPRE) on the promoter of PPAR target genes; the D structural domain, which is the docking site for various cofactors; and the large Y-shaped hydrophobic binding pocket formed at the C-terminus, which is the ligand-binding structural domain (E/F structural domain) that confers the ability to bind endogenous or exogenous lipophilic ligands to PPARs
- Lines 652-657
Chiglitazar Sodium is the first PPAR full agonist independently developed in China and is approved for the treatment of T2DM. It was included in the national health insurance program in January 2023. Owing to the simultaneous moderate activation of three functionally different but overlapping PPAR subtypes, Chiglitazar Sodium attains the ability to not only selectively alter the expression of a series of genes related to insulin sensitivity
- Lines 660-664
The results of phase III clinical trials of Chiglitazar Sodium showed that it significantly reduced HbA1c levels by 1.52%, and its hypoglycemic efficacy was stable for up to 52 weeks, with improvements in fasting and 2-h postprandial glucose compared with that with sitagliptin. It also significantly improved insulin sensitivity, protected islet secretion, and reduced triglyceride levels.
- Lines 674-680
T2DM are mainly conditionally pathogenic, such as Bacteroides caccae, Escherichia coli, and Eggerthella lenta, and they reduce abundance of butyrate-producing bacteria, including Eubacterium rectale, Faecalibacterium prausnitzii, and Roseburia intestinalis [131]. Although the specific mechanisms by which gut microbiota affect the development of T2DM are complex, some of the clearer mechanisms involve the short-chain fatty acid (SCFA), bile acid (BA), branched-chain amino acid (BCAA), endotoxin-intestinal barrier, imidazole propionate (ImP), and aryl hydrocarbon receptor (AhR) theories
- Lines 690-693
Butyrate can bind to GPR41 and GPR43 to protect β-cells from oxidative stress, inhibit deacetylation of histones, down-regulate the expression of genes related to gluconeogenesis in the liver, and bind to GLP-1R to promote insulin secretion
- Lines 695-702
The main gut microbiota are involved in the synthesis, modification, and signal transduction of BAs. These microbiota have the ability to produce bile salt hydrolases, enzymes that convert primary BAs into secondary BAs. This process is crucial for the biological functions of BAs [139]. Primary BAs can bind to and activate the farnesoid X receptor (FXR), which improves glucose tolerance and increases insulin sensitivity [134,138]. Secondary BAs bind to Takeda-G-protein receptor-5 (TGR-5) and induce GLP-1 secretion, thereby promoting insulin secretion [129].
- Lines 712-715
LPS is recognized by innate toll-like receptors, which later activate the nuclear factor kappa-B (NF-κB) signaling pathway, promoting the release of pro-inflammatory factors, thereby leading to inflammation. This in turn triggers β-cell destruction, insulin resistance, and increased intestinal wall permeability [130,134].
- Lines 720-723
ImP is a histidine-derived metabolite elevated in prediabetic and T2DM patients [132], and has been shown to impair insulin signaling by activating mTORC1 and in-creasing Ser phosphorylation of IRS-1 [142]. Furthermore, it impairs the effects of metformin by inhibiting AMPK activity [143].
- Lines 733-735
Mixtures of probiotics and prebiotics or combinations of both with antidiabetic drugs have a superior efficacy than probiotics or prebiotics alone [144,145].
- Lines 738-739
However, there are safety concerns; one case of severe bacteremia and one death have been reported.
- Lines 749-751
Cells can release miRNAs in free form or in complex with extracellular vesicles, which can be taken up by other types of cells, thus mediating cell-to-cell actions [148].
- Lines 752-755
furthermore, they influence insulin secretion by regulating β cell metabolic stress and proliferation, survival, as well as regulating GSIS and improving insulin sensitivity [148-151]. Therefore, miRNAs are potential biomarkers for diabetes prediction [152].
- Lines 759-763
However, there is heterogeneity in the expression of miRNAs, and some studies have found sex variability in their use as markers, which may be owing to the following mechanisms: estrogen regulates the transcription and processing of miRNAs, incomplete X-chromosome inactivation leading to biallelic expression of miRNAs, and regulation of miRNA expression by epigenetics [154].
- Lines 777-778
Additionally, further knockdown experiments confirmed that GI plays a key role [156].
- Lines 782-787
However, recent findings have suggested that ChREBP is required for glucose-stimulated β-cell proliferation. Overexpression of ChREBPβ leads to glucose toxicity and its subsequent death of β-cells, while overexpression of ChREBPα enhances glucose-stimulated β-cell proliferation as it stimulates the Nrf2 antioxidant pathway, thereby preventing oxidative damage [157]. Therefore, the development of ChREBPα selective activators are potential treatment against diabetes.
- Lines 789-791
Microexons are DNA sequences that encode proteins, approximately 3 to 27 nucleotides long, which can be selectively spliced in neurons, microglia, embryonic stem cells, and cancer cells to produce cell type-specific protein isoforms [158].
- Lines 794-798
If depletion of SRRM3 in human and rat β-cell lines, as well as mouse pancreatic islets, or the use of antisense oligonucleotides to inhibit specific IsletMICs, can result in in-appropriate insulin secretion, it suggests that IsletMICs are present at low levels in patients with T2DM; hence, up-regulation of IsletMICs levels is a potential therapeutic strategy [159].
- Lines 800-807
Although there is a genetic predisposition to T2DM, the onset and progression of the disease are influenced more by the environment, with unhealthy diet and lack of exercise being major factors. Despite continuous attempts to develop new drugs, side effects remain challenging, and even the first-line drug, metformin, can cause nausea and gastric distension. There is also the issue of compliance with medication, and symptoms may not improve once the medication is stopped. Therefore, intervention through good lifestyle habits such as proper diet and exercise, with medication as adjunctive therapy, will be a lower-cost and healthier treatment option.
- Lines 808-810
Diet is a cornerstone in the prevention and treatment of abnormal glucose metabolism and insulin resistance [160]. The ADA released an updated consensus report on diabetes nutrition therapy in 2019 [161].
- Lines 813-815
but regardless of the pattern, it is generally based on three basic principles, as follows: consuming more non-starchy vegetables (such as broccoli, kale and mushrooms), less sugar and refined grains, and choosing natural foods over highly processed foods.
- Lines 831-833
Besides, the Mediterranean diet, low-sugar diet, and medium-carbohydrate diet have shown better control of HbA1c and fasting blood glucose levels in patients [170].
- Lines 835-846
while polyphenols have anti-inflammatory and antioxidant properties, reduce β-cell apoptosis, inhibit α-glucosidase, modulate the intestinal microbiota, and improve adipose tissue metabolism [172]. While ω-3 appears to have different efficacy, a previous large-scale prospective study showed that ω-3 reduces the risk of T2DM [173]; however, a recent mendelian randomized study showed that ω-3 can act through two different clusters of genetic variants, with one set of variants ameliorating insulin resistance while the other contributing to β-cell dysfunction and increasing T2DM risk [174]. It is believed that potentially harmful diets should be completely eliminated and beneficial diets should be completely followed, but diets have complexities. Excessive fiber intake can lead to belly bloating and diarrhea; abstainers have a higher prevalence of the disease than moderate drinkers; and indiscriminate combinations and large stacks of macronutrients are counterproductive [161].
- Lines 856-858
Improvements in systemic insulin sensitivity in patients can last from 2 to 72 h after exercise and the reduction in blood glucose is closely related to the duration and in-tensity of exercise.
- Lines 861-867
Breaking up sedentary behavior by doing "little and often" daily physical activity can moderately lower postprandial blood glucose levels, especially in patients with insulin resistance and high BMI. Exercise safety should also be emphasized. In cases where blood glucose levels exceed 250 mg/dL, exercise should be prohibited. Additionally, patients who are on insulin or insulin secretagogues should have fast-acting carbohydrates, such as candy, readily available to prevent hypoglycemia [178].
- Lines 871-873
Diabetes is a public health issue of global concern as a chronic multifaceted disease that remains incurable and is accompanied by many complications such as cardiovascular disease and emotional-cognitive dysfunction.
- Lines 874-876
Concurrently, owing to the large population base, the prevalence of diabetes in China will only increase as the population aging process continues to accelerate.
- Lines 880-882
such as Semaglutide, which has weight loss, hypoglycemic, and cardiovascular protective effects, and as a long-acting drug, it can be injected once a week, eliminating the tediousness of daily dosing.
- Lines 883-884
Stem cell therapy is also making its debut in the field of diabetes.
- Lines 888-893
Despite the continuously proposed treatment strategies, China still has the highest number of patients with diabetes in the world. Ultimately, individual differences, diversity of pathogenesis, and drug resistance have a significant influence. Different patients may have different underlying diseases, and drugs targeting a certain target may not be suitable for everyone. For instance, metformin is not suitable for patients with heart failure or renal dysfunction.
- Lines 899-900
Therefore, future studies must investigate the changes in the omics of lesioned cells and the specific metabolic pathways they affect.
- Lines 902-904
In conclusion, the treatment and prevention of diabetes requires a concerted effort by all parties involved and not just scientist/physicians.
- - Line 826 keto acidosis could occur in type I diabetes, but not in type II if the diet is well designed. The part on KD could be a bit expanded, highlighting the anti-inflammatory and antioxidant action of this kind of diet (10.3390/antiox8080269 and 10.1016/j.eplepsyres.2020.106454 for example)
Response:
We are very grateful to the reviewers for their valuable comments. We have added content on the anti-inflammatory and antioxidant effects of KD in the new manuscript (lines 822-831). The details are as follows:
Furthermore, it has been observed that diabetes leads to dysregulation in the anti-inflammatory and oxidative stress pathways. However, the implementation of a KD has shown potential in mitigating these detrimental effects. The KD exerts its anti-inflammatory effects through various mechanisms such as adenosine, ketone bodies, mTOR pathways, PPARγ, NLRP3 inflammasome, and gut microbiota. Additionally, the KD exhibits antioxidant properties by improving mitochondrial dynamics, regulating miRNA expression associated with oxidative stress, and enhancing the pentose phosphate pathway and related antioxidant defense systems [165-168]. However, it should be noted that a specific ketogenic diet plan may lead to metabolic ketoacidosis if it is not well designed [169].
Reference:
- Koh, S.; Dupuis, N.; Auvin, S. Ketogenic diet and Neuroinflammation. Epilepsy Res. 2020, 167, 106454. DOI:10.1016/j.eplepsyres.2020.106454.
- Paoli, A.; Cerullo, G. Investigating the Link between Ketogenic Diet, NAFLD, Mitochondria, and Oxidative Stress: A Narrative Review. Antioxidants (Basel). 2023, 12, 1065. DOI:10.3390/antiox12051065.
- Cannataro, R.; Caroleo, M.C.; Fazio, A.; La Torre, C.; Plastina, P.; Gallelli, L.; Lauria, G.; Cione, E. Ketogenic Diet and microRNAs Linked to Antioxidant Biochemical Homeostasis. Antioxidants(Basel). 2019, 8, 269. DOI:10.3390/antiox8080269.
- Peng, F.; Zhang, Y.-H.; Zhang, L.; Yang, M.; Chen, C.; Yu, H.; Li, T. Ketogenic diet attenuates post-cardiac arrest brain injury by upregulation of pentose phosphate pathway-mediated antioxidant defense in a mouse model of cardiac arrest. Nutrition. 2022, 103–104, 111814. DOI:10.1016/j.nut.2022.111814.
- Churuangsuk, C.; Hall, J.; Reynolds, A.; Griffin, S.J.; Combet, E.; Lean, M.E.J. Diets for weight management in adults with type 2 diabetes: an umbrella review of published meta-analyses and systematic review of trials of diets for diabetes remission. Diabetologia. 2022, 65, 14–36. DOI:10.1007/s00125-021-05577-2.
